# Using a composite flow law to model deformation in the NEEM deep ice core, Greenland: Part 2 the role of grain size and premelting on ice deformation at high homologous temperature

Ernst-Jan N. Kuiper[1,2], Johannes H. P. de Bresser[1], Martyn R. Drury[1], Jan Eichler[2,3], Gill M. Pennock[1], Ilka Weikusat[2,1,3]

[1]Faculty of Earth Science, Utrecht University, 3508 TA Utrecht, the Netherlands
[2]Alfred Wegener Institute, Helmholtz Centre for Polar and Marine Research, 27570 Bremerhaven, Germany
[3]Department of Geosciences, Eberhard Karls University Tübingen, 72074 Tübingen, Germany

*Correspondence to*: M.R.Drury@uu.nl and Ilka.Weikusat@awi.de

**Abstract.** The ice microstructure in the lower part of the North Greenland Eemian Ice Drilling (NEEM) ice core consists of relatively fine grained ice with a single maximum crystallographic preferred orientation (CPO) alternated by much coarser grained ice with a partial (great circle) girdle or multi-maxima CPO. In this study, the grain size sensitive (GSS) composite flow law of Goldsby and Kohlstedt (2001) was used to study the effects of grain size and premelting (liquid-like layer along the grain boundaries) on strain rate in the lower part of the NEEM ice core. The results show that the strain rates predicted in the fine grained layers are about an order of magnitude higher than in the much coarser grained layers. The dominant deformation mechanisms, based on the flow relation of Goldsby and Kohlstedt (2001), between the layers is also different with basal slip rate limited by grain boundary sliding (GBS-limited creep) being the dominant deformation mechanism in the finer grained layers, while GBS-limited creep and dislocation creep (basal slip rate limited by non-basal slip) contribute both roughly equally to bulk strain in the coarse grained layers. Due to the large difference in microstructure between finer grained ice and the coarse grained ice at premelting temperatures (T>262K), it is expected that the fine grained layers deform at high strain rates, while the coarse grained layers are relatively stagnant. The difference in microstructure, and consequently in viscosity, between impurity-rich and low impurity ice can have important consequences for ice dynamics close to the bedrock.

## 1 Introduction

As a consequence of anthropogenic global warming, it is expected that global mean sea level (GMSL) rise will accelerate in the next decades and centuries (e.g. IPCC, 2014; Kopp et al., 2017). To improve the predictions in GMSL rise, the flow of ice in polar ice sheet models should be accurately described, since it is expected that the melting of the polar ice sheets will contribute significantly to future GMSL rise (IPCC, 2014). Therefore, a full understanding and description of ice deformation mechanisms in polar ice sheets is required. Ice in the lower part of polar ice sheets is of particular interest. Generally, this ice is expected to deform much faster than the ice closer to the surface, due to the relatively high temperatures and shear stress increase towards the bedrock. However, stress and strain rates near the bedrock can vary considerably due to the influence of variations in bedrock topography, slippery patches or subglacial lakes (e.g. Budd and Rowden-rich, 1985; Gudlaugsson et al., 2016; Wolovick and Creyts, 2016). The high variation in deformation rates in the ice near the bedrock are often shown by borehole surveys (e.g. Gow and Williamson, 1976; Paterson, 1983; Morgan et al., 1998; Thorsteinsson et al., 1999; Weikusat et al., 2017). The main source of heat of the ice near the bedrock is geothermal heat, which typically has a strength of 50-100 mW m$^{-2}$ (Rogozhina et al., 2016). Locally, other sources of heat can be important, like strain heating (e.g. Krabbendam, 2016) or latent heat released by refreezing of melt water (e.g. Zwally et al., 2002; Van de Wal et al., 2008; Catania and Neumann, 2010).

Ice deformation experiments have shown that the effect of temperature on the strain rate of ice can be described via a calibrated Arrhenius type flow law relating equivalent strain rate to equivalent stress and temperature (e.g. Glen, 1955; Homer and Glen, 1978). However, at temperatures close to the melting point, the strain rate increase has been found to be higher than predicted by the extrapolation of lower temperature results using the Arrhenius relation (e.g. Budd and Jacka, 1989), inferred to be due to the initiation of a new high temperature deformation mechanism or due to the enhancement of the orginal one. To account for the latter, a higher activation energy is often used above a certain temperature threshold (e.g., Mellor and Testa, 1969; Barnes et al., 1971; Morgan, 1991; Paterson, 1994; Goldsby and Kohlstedt, 2001). This higher activation energy, which is the result of a simple best-fit approach of temperature and strain rate using a Arrhenius type relation, lacks a good link with the controlling deformation mechanism, assumed to be intracrystalline dislocation motion. Therefore, its application is well established, but is not well founded for two reasons. First, the enhancement in strain rate in the high temperature regime is not described well by a constant activation energy (Mellor and Testa, 1969; Morgan, 1991; Budd and Jacka, 1989; Treverrow et al., 2012). In fact the apparent activation energy for secondary (or minimum) creep rate changes from 77 to 500 kJ/mol in the temperature range from -10 to -0.05 °C (Budd and Jacka, 1989), so the assumption of a constant value is a gross simplification. Second, there is no particular microphysical basis for an increased activation energy for creep in the high temperature regime. The enhanced creep rate may be caused by an effect of brine content on creep rate, with brine content increasing with temperature. In experiments on ice single crystals and bicrystals close to the melting point (Homer and Glen, 1978; Jones and Brunet, 1978), a strong strain rate increase has not been observed. This strongly suggests that the strain rate enhancement in natural ice close to the melting point is related to the occurrence of grain boundaries.

Premelting along grain boundaries has often been stated as being responsible for the high strain rates during ice deformation tests at high temperatures (e.g. Mellor and Testa, 1969; Dash et al., 1995, 2006; Wilson et al., 1996; Wilson and Zhang, 1996). Premelting is expected to initiate at surfaces or interfaces such as grain boundaries where the crystallographic structure is disrupted and molecules can adopt liquid-like qualities at temperatures below the melting point (e.g. Orem and Adamson, 1969; Barnes et al., 1971; Dash et al., 1995, 2006; Döppenschmidt et al., 1998). A vast amount of theory and indirect measurements indicate that quasi-liquid, disordered layers are present along ice surfaces and grain boundaries at 10 to 20°C below the bulk melting temperature (Dash et al. 2006; Vaughan et al. 2006; Thomson et al., 2013). At temperatures above -15 to -10° C grain boundary mobility in ice increases by two to four orders of magnitudes (Duval and Castelnau, 1995; Schulson and Duval, 2009) and premelting may be responsible for this increase of grain boundary mobility (De la Chapelle et al. 1998). In this paper we accept the hypothesis that the enhanced creep rate and grain boundary mobility in ice at temperatures above about -10° C are related to premelting along grain boundaries, although as noted by Dash et al. (2006) "it has yet to be conclusively demonstrated whether or not grain boundary melting contributes to the explanation for these phenomena".

The different activation energy reflecting the onset of premelting is used widely at a temperature threshold of -10°C (263K) when applying Glen's flow law in ice sheet models (e.g. Paterson, 1994), but a higher temperature threshold of -8°C (285K) (Barnes et al., 1971) or lower temperature thresholds of -15°C (258K) to -18°C (255K) (Goldsby and Kohlstedt, 2001) should be considered. As noted above, the assumption of a constant activation energy above the temperature threshold, is a gross simplification, however as this approach is widely used in ice sheet models and the only available grain size sensitive flow laws (Goldsby and Kohlstedt, 2001) use a constant activation energy in the high temperature regime, we will follow this approach. As temperature generally increases with depth in polar ice sheets, the temperature where premelting starts to affect ice rheology is important. A low premelting temperature would mean that a considerable larger portion of the ice is affected by premelting, hence will show strain rate enhancement over a larger range, compared to a higher temperature threshold.

At temperatures close to the melting point in the lower part of polar ice cores, a particular type of microstructure with very coarse grains and interlocking grain boundaries is often found (e.g. Gow and Williams, 1976; Gow and Engelhardt, 2000; NEEM community members, 2013; Weikusat et al., 2017). In the lower part of the North Greenland Eemian Ice Drilling (NEEM) ice core (NEEM community members, 2013) such regions of coarse, interlocking grains are alternated by regions of

finer grains. In case of the NEEM ice core, the alternated layers of fine and coarse grained ice are the results of stratigraphic disruptions and overturned folds, which consist of low impurity Eemian interglacial ice and impurity-rich ice from the late Eemain and from the penultimate glacial period (Figure 1) (NEEM community members, 2013).

These coarse grained layers at the bottom of some polar ice cores typically possess crystallographic preferred orientations (CPOs) that could be described as a multi maxima, although the exact type of CPO is often unclear due to the low number of grains measured in individual thin sections. The coarse grains and multi maxima CPO are thought to be the result of rapid strain induced boundary migration (SIBM) in combination with the nucleation of new grains (SIBM-N) (e.g. Alley, 1992; Duval and Castelnau, 1995; Durand et al., 2009; Faria et al., 2014a), where based on the coarse grain size SIBM is likely more important than nucleation of new grains. SIBM is the migration of grain boundaries driven by the difference in stored strain energy between neighbouring grains that result from lattice distortions such as dislocations (Humphreys and Hatherly, 2004). SIBM starts in firn (Kipfstuhl et al., 2006, 2009) and is assumed to increase grain size (e.g. Duval and Castelnau, 1995), although under certain conditions SIBM can also be a grain size reducing mechanism by grain dissection (Steinbach et al., 2017). The layers of alternating grain size in deeper NEEM samples allows for the study of the effect of grain size on strain rate and CPO development, in the high temperature regime where enhanced creep and recrystallization can be explained by the occurrence of premelting along grain boundaries.

In this study, the composite flow law of Goldsby and Kohlstedt (2001) was used to explore the effect of grain size on strain rate in the lower part of the NEEM ice core. The composite flow law describes the deformation of polycrystalline ice as a combination of grain size sensitive (GSS) and grain size insensitive (GSI) deformation mechanisms, where the dominant deformation mechanism depends on the temperature, stress and grain size. Compared to our companion paper (part 1), which describes ice deformation at relatively low temperatures in the upper 2207 m of the NEEM ice core, the emphasis in this paper is on possible differences in the controlling deformation mechanism close to the melting point. The results from flow law calculations were combined with CPO data to study deformation mechanisms in the lower part of the NEEM ice core. The NEEM ice core was chosen because of the high density of reflective light microscopy (LM) images (Kipfstuhl, 2010; Binder et al., 2013) and the alternation of fine and coarse grained layers in the lower part of the ice core. These reflective LM images were used to determine the change in grain size with depth (Binder, 2014). Furthermore, the CPO data of the NEEM ice core (Eichler et al., 2013) were available. A critical assessment of the temperature threshold for the onset of enhanced recrystallization and creep, caused by premelting, in polar ice sheets led to a modification of the composite flow law of Goldsby and Kohlstedt (2001). The NEEM ice core data were used to calculate the strain rate predicted by Glen's flow law (Glen, 1952, 1955; Paterson, 1994) and by the modified composite flow law of Goldsby and Kohlstedt (2001) in the lower part of the NEEM ice core. Comparing the results from the modified composite flow law with the results obtained using Glen's flow law is an interesting 'control' since Glen's flow law is the most often used flow law in polar ice sheet models. Possible deformation mechanisms are discussed.

## 2 Methods

### 2.1 The NEEM ice core and ice microstructure data

The NEEM ice core, located in the northwest of Greenland (77.45°N, 51.06°W), was drilled during the field seasons of 2008-2012 (NEEM community members, 2013; Rasmussen et al., 2013). The NEEM ice core is 2540 m long with ice deposited during the Holocene (Holocene ice) extending to a depth of 1419 m (Montagnat et al., 2014). Ice deposited during the last glacial period (glacial ice) lies below reaching a depth of 2207 m. The layering in the underlaying Eemian ice is strongly disturbed and four stratigraphic disruptions were identified at 2209.6 m, 2262.2 m, 2364.5 m and 2432.2 m of depth by discontinuities in oxygen stable isotope values ($\delta^{18}O_{ice}$) of $H_2O$ and methane ($CH_4$) concentrations (NEEM community members, 2013). These repeated and overturned parts are mainly from the late Eemian, which causes the coarse grained Eemian

ice to be alternated with the finer grained late Eemian ice. Below 2432.2 m of depth, in the lowest 330 m of the NEEM ice core there is undated finer grained, impurity rich ice that is thought to be from the penultimate glacial. In the remainder of this study, the ice from 2207 m of depth to the ice-bedrock interface at 2540 m of depth will be referred to as the 'Eemian-glacial facies'.

The lower part of the glacial ice of the NEEM ice core, from 2000 to 2207 m of depth was included in this study to show the contrast in grain size, CPO and calculated strain rate between the glacial ice and the deeper Eemian-glacial facies. Therefore, the available LM images, orientation images and CPO data below 2000 m of depth were included in this study. Calculations using the GSS composite flow law of Goldsby and Kohlstedt require grain size as an input variable. The grain size data in the lowest 540 m of the NEEM ice core were obtained using 224 large area scanning macroscope (LASM) images

(Kipfstuhl, 2010) taken using reflective light macroscopy (Krischke et al., 2015). This method uses thermal etching by sublimation to reveal (sub)grain boundaries as grooves on the surface of the ice core sample (e.g. Saylor and Rohrer, 1999) and has a resolution of 5 µm per pixel edge. Each LASM image is about 90 mm long by about 55 mm wide (Kipfstuhl et al., 2006) and was digitally analyzed using the Ice-image software (www.ice-image.org) (Binder et al., 2013; Binder, 2014). The software automatically detects the grain area of each grain by counting the pixels enclosed by grain boundaries. The total area

classified as 'grain' by the Ice-image software was divided by the number of grains to give a mean area. The equivalent grain diameter was calculated from the mean area; this is always larger than the mean diameter calculated from the population of diameter measurements. Deriving a full grain size distribution, as was done in the companion paper (part 1), was not possible in the Eemian-glacial facies because the number of grains in the LASM images (90 x 55 mm) of the coarse grained layers was too low. Grains with a diameter <0.3 mm were excluded from the data set as these grains are often artefacts caused by relaxation

(Binder, 2014). In some parts of the Eemian-glacial facies, the grains are up to several centimetres and the grain boundaries are irregular with many grain boundary bulges. It is not possible to determine a mean grain size for these ice core sections by the Ice-image software as these grains often cross the edge of the LM images and are therefore not included in the grain size data (Binder, 2014). After visual inspection of the LASM images to check the grain size, the ice core sections containing very large grains were assigned a mean grain diameter of 30 mm, which was the estimated grain diameter based on the LASM

images.

     An automatic Fabric Analyzer (FA) was used to create high-resolution maps of the c-axes orientations. The method is based on the double-refracting properties of the hexagonal ice crystal. It offers a fast alternative for the measurement of crystal orientations, which are, however, limited only to the c-axes. The grains in each orientation image were plotted in a pole figure. For each orientation images the Woodcock parameter, $k$, was calculated according to (Woodcock, 1977):

$$k = \frac{\ln (\lambda_3/\lambda_2)}{\ln (\lambda_2/\lambda_1)}, \hspace{3cm} (1)$$

where $\lambda_1$, $\lambda_2$ and $\lambda_3$ are the normalized eigenvalues of the second order orientation tensor with $\lambda_1 < \lambda_2 < \lambda_3$. The Woodcock parameter is often used in order to distinguish between cluster and girdle type of CPOs. A distribution with a Woodcock parameter >1 indicates a cluster, while a Woodcock parameter <1 indicates a girdle. Further computer-based analysis (Eichler et al., 2013) of the FA-CPO-maps enables the derivation of a variety of microstructural parameters, such as mean grain shape

or size. These can be used as complementary values to the LASM analysis data. However, these microstructural parameters can differ significantly from the microstructural parameters obtained by the Ice-image software (www.ice-image.org) (Binder et al., 2013; Binder, 2014). For instance, the effective diameter derived from the orientation images in this study is systematically shifted towards lower values, which is mainly caused by the exclusion of grains with a grain diameter <0.3 mm in the LASM method as was described above. In the remainder of this paper, the grain size (effective diameter) determined

using the Ice-image software (Binder et al., 2013; Binder, 2014) will be used unless stated otherwise.

## 2.2 Flow laws and flow law parameters

One of the two flow laws that was used during this study is the composite flow law of Goldsby and Kohlstedt (2001). The composite flow law was derived during uniaxial deformation tests in secondary creep with very fine grained ice (Goldsby and Kohlstedt, 1997, 2001). The composite flow law combines different deformation mechanisms of ice explicitly, instead of presenting a series of individual flow laws. During their uniaxial deformation experiments on artificial fine grained ice in secondary creep, Goldsby and Kohlstedt (1997, 2001) defined a composite flow law with rate contributions from four mechanisms:

$$\dot{\varepsilon} = \dot{\varepsilon}_{disl} + \left(\frac{1}{\dot{\varepsilon}_{basal}} + \frac{1}{\dot{\varepsilon}_{GBS}}\right)^{-1} + \dot{\varepsilon}_{diff}, \qquad (2)$$

where $\dot{\varepsilon}$ is the strain rate. $\dot{\varepsilon}_{disl}$ refers to dislocation creep where basal slip is the main strain producing mechanism which is rate limited by non-basal slip and $\dot{\varepsilon}_{diff}$ refers to diffusion creep. Basal slip and grain boundary sliding are sequential processes acting together, where the slower mechanism determines the overall strain rate by rate limiting the faster mechanism (Durham and Stern, 2001). In the part in between brackets in equation (2), $\dot{\varepsilon}_{basal}$ refers to basal slip, while $\dot{\varepsilon}_{GBS}$ refers to GBS. Under stress, grain size and temperature conditions appropriate for terrestrial ice, diffusion creep and grain boundary sliding rate limited by basal slip are not relevant for ice deformation (Goldsby and Kohlstedt, 2001; Goldsby, 2006). Therefore, the composite flow law simplifies to:

$$\dot{\varepsilon} = \dot{\varepsilon}_{disl} + \dot{\varepsilon}_{GBS}, \qquad (3)$$

The strain rate produced by creep can be described by the following general flow law:

$$\dot{\varepsilon} = A\,\sigma^n\,d^{-p}\,\exp\left(-\frac{Q}{R\,T}\right), \qquad (4)$$

where $\dot{\varepsilon}$ is the strain rate (s$^{-1}$), A is a material parameter, $\sigma$ is the stress (MPa), n is the stress exponent (dimensionless), $d$ is the grain size (m), $p$ is the grain size exponent (dimensionless), Q is the activation energy (kJ mol$^{-1}$), R the gas constant (J K$^{-1}$ mol$^{-1}$) and T the absolute temperature (K) corrected for the change in pressure-melting point due to the cryostatic pressure. The value for p determines whether the creep is grain size insensitive (p=0) or grain size sensitive (p≠0). The mean grain diameter, as determined from LASM images with the Ice-image software, was used for the calculation of the strain rate produced by GBS-limited creep.

If the temperature is expressed in terms of the difference with the pressure-melting point, ice can, as a first order approximation, be considered incompressible (Rigsby, 1958; Doake and Wolff, 1985, Greve et al., 2014). The pressure-melting temperature was calculated according to:

$$T_m = -C\,\Delta P, \qquad (5)$$

where $T_m$ is the pressure-melting temperature (°C), $C$ is the pressure-melting constant for glacier ice ($9.8\ 10^{-8}\ °C\ Pa^{-1}$; Lliboutry, 1976) and $\Delta P$ is the overburden pressure (Pa) which was calculated according to:

$$\Delta P = \rho_{ice}\,h_{ice}\,g, \qquad (6)$$

where $\rho_{ice}$ is the density of ice (910 kg m$^{-3}$), $h_{ice}$ is the ice thickness (m) and $g$ is the gravitational constant (9.81 m s$^{-2}$). The in-situ temperature ($T$) profile of the NEEM ice core was taken from Sheldon et al. (2014). In the remainder of this paper, $T^*$ is used for the difference in temperature of the ice at a certain depth with respect to the pressure-melting point at that depth, which can be calculated according to

$$T^* = T - T_m. \qquad (7)$$

$T^*$ was used in Equation (4) to calculate the strain rates in the lower part of the NEEM ice core according to Glen's flow law and the composite flow law. At the base of the 2540 m long NEEM ice core the pressure-melting point is calculated to be -2.2°C (271K). For example, the in-situ temperature at the ice-bedrock interface at NEEM is -3.4°C, which would give a $T^*$ of -1.2°C (Equation 7).

The other flow law that was used in the this study is "Glen's flow law" (Glen, 1952, 1955; Paterson, 1994), which is a group of flow laws that are commonly used in ice sheet modelling. Glen's flow law is based on the results of uniaxial

compression experiments on artificial, (initially) isotropic polycrystalline ice. Glen's flow law has the same form as Equation (4), but is independent of grain size (i.e., p=0) and the form of Glen's flow law that is most often used has a stress exponent of n=3 (Paterson, 1994). However, at the low stress conditions with strong CPO development, a higher value of n of 3,5-4 might be a better approximation of ice dynamics in polar ice sheets as was shown by Treverrow et al. (2012), Bons et al. (2018) and others.

Following the analysis presented in our companion paper (part 1), a constant equivalent stress of 0.07 MPa was taken as input for Glen's flow law and the composite flow law. This assumption is a useful first approximation for the NEEM ice core where the equivalent stress, related to the shear stress in the lower part of the ice core, is by coincidence similar to the magnitude of equivalent stress related to the vertical stress in the upper part of the ice core. We explore the effect of grain size on the dominant deformation mechanism and the total strain rate, it is beyond the scope of this study to derive a stress-depth model for NEEM because this requires knowledge on the rheology, which is the property that is investigated here. When calculating strain rates, no distinction is made between simple shear and pure shear deformation.

## 3 Results

### 3.1 Ice microstructure in the Eemian-glacial facies

Figure 2 shows reflective LM images and orientation images with pole figures of three ice core sections at 2258 m, 2264 m, 2308 m of depth in the Eemian-glacial facies. Only a part of the 90 x 55 mm ice core sections is shown in the LM and fabric images, while the pole figure shows the c-axes of all the grains in the 90 x 55 mm ice core section. The ice core section at 2258 m of depth in Figure 2a is from just above the stratigraphic disruption at 2262.2 m of depth, while the ice core section at 2264 m of depth in Figure 2b is from just below this stratigraphic disruption. The ice core section in Figure 2c is from 2308 m of depth in the middle of one of the overturned layers (NEEM community members, 2013). The ice core section in Figure 2a is one of the finer grained ice core sections in the Eemian-glacial facies with a mean grain size of about 5 mm and originates from the late Eemian (NEEM community members, 2013). The orientation image and pole figure of Figure 2a show that, similar to the glacial ice (1419-2207 m of depth; Montagnat et al., 2014), almost all c-axes are strongly aligned parallel to the vertical ice core axis. However, compared to the glacial ice, the grain size of the ice core section in Figure 2a is slightly larger and the grain shape is more irregular. The ice core section in Figure 2b is one of the ice core sections that was given a mean grain size of 30 mm, typical of the Eemian ice with high $\delta^{18}O_{ice}$ and low impurity content deposited unders warm interglcial conditions (NEEM community members, 2013). The grains have an irregular shape with many millimetre sized bulges along the grain boundaries. The orientation image and pole figure show that the c-axes are distributed in a partial (great circle) girdle spanning about 40° from the vertical axis. Although in some cases the c-axes distribution can also be classified as a multi maxima CPO. The ice core section in Figure 2c has a mean grain size of about 7 mm and originates from the end of the Eemian period (NEEM community members, 2013). The grain boundaries are bulging and have an irregular shape. The orientation image and pole figure show that the c-axes are distributed in a partial (great circle) girdle spanning about 30° to 40° from the vertical axis.

### 3.2 Correlation mean grain diameter with type and strength of CPO

The correlation between the mean grain diameter and the type and strength of CPO in the lowest 540 m of the NEEM ice core is shown in Figure 3a and 3b. This figure shows the first c-axes eigenvalue, $\lambda_3$ in equation (1) (Eichler et al., 2013) and the Woodcock parameter (Woodcock, 1977) versus the mean grain diameter for each orientation image in the lower part of the glacial ice (2000-2207 m of depth) and the ice in the Eemian-glacial facies (2207-2540 m of depth). The ice from the lower part of the glacial period (2000-2207 m of depth), which has a finer mean grain diameter (about 2 mm) than the ice in the

Eemian-glacial facies, has a c-axes eigenvalue of $\lambda_3 > 0.9$ and the Woodcock parameter typically varies from 4-10. Some of the finest grained regions in the Eemian-glacial facies have a slightly larger mean grain diameter (3-5 mm) with a similar c-axes eigenvalue of $\lambda_3 > 0.9$ and a rather similar Woodcock parameter of about 2-10. For ice core sections with a mean grain diameter larger than about 5 mm, the eigenvalue is $\lambda_3 < 0.9$ with a Woodcock parameter varying from 0.3-10. However, the number of grains per orientation image decreases with increasing grain size and therefore the statistical significance of the first eigenvalue and Woodcock parameter decreases with increasing mean grain diameter. Based on Figure 3, two classes of microstructure can be distinguished based on mean grain diameter and type and strength of CPO. The first class has a relatively fine mean grain diameter of <5 mm that is comparable in eigenvalue and Woodcock parameter to the glacial ice (green rectangle). The other class has a mean grain diameter of >5 mm with a relatively low first eigenvalue and Woodcock parameter (yellow rectangle). The mean grain diameter in Figure 3 was derived from orientation images, which gives a slightly lower mean grain diameter than the mean grain diameter derived using the Ice-image software, like Figure 2.

**3.3 Transition temperature: NEEM results compared to other polar ice cores**

Table 1 shows data from eight polar ice cores drilled at the Greenland and Antarctic ice sheets that contain a sudden increase in grain size and change in CPO in the lower part of the ice core. The in-situ temperature at the bottom of the boreholes varies significantly between the ice cores. The ice near the bedrock at GISP2 and GRIP was frozen to the bed, while for Byrd, EDC, EDML, NEEM and Siple dome the ice was at, or very close to, pressure-melting point. In all eight ice cores, the CPO and grain size start to change at an in-situ temperature of about -13°C (260K). For the NEEM ice core, this transition coincides with the climatic transition of the end of the Eemian period, also known as marine isotope stage (MIS) 5e, to the beginning of last glacial period (MIS 5d). The transitions to coarse grains with a multi maxima CPO in the EDML, GISP2 and GRIP ice core also coincides with a climatic transition. For the other four ice cores, the transition to a different microstructure does not coincide with a major climatic transition. The pressure corrected temperature threshold (T*, Equation 5-7) at which this transition occurs in these polar ice cores is remarkably constant at T* of -11°C (262K). The depth in the ice cores at which the microstructure changes to large grains and a multi maxima CPO is different for each ice core, which leads to a slightly different T* when correcting for the change in pressure-melting point with depth (Equation 5-7). However, when taking into account the uncertainty in determining the in-situ temperature at a certain depth and the possible influence of the different impurity content between the ice cores and individual layers in the same ice core that affect the pressure-melting point, the effect of correcting the temperature at which the microstructure changes due to changes in overburden pressure is small. A different cause for the sudden change in ice microstructure is stress relaxation due to the disruption of simple shear flow by the bedrock topography. This leads to more stagnant ice which, at the elevated temperatures close to the bedrock, can start to recover and recrystallize. However, the alternation of different microstructures with finer and coarser grained as found at NEEM cannot be explained in this way.

The flow law parameters of the original composite flow law (Goldsby and Kohlstedt, 2001) shown in Table 2 were modified to fit the temperature (T*) threshold of -11°C (262K) for premelting as derived from Table 1 and the experimental data points from Goldsby and Kohlstedt (2001). Another reason for modifying the dislocation creep parameters is improve the fit of the dislocation creep flow law and the experimental data of Goldsby and Kohlstedt (2001) as explained in the companion paper (part 1). Both the flow law parameters for dislocation creep and GBS-limited creep were modified and are given in Table 3. Since the flow law parameters for GBS-limited creep were adjusted to be consistent with a temperature (T*) threshold of 262K, they are different from the flow law parameters used in our companion paper.

Figure 4 shows the temperature versus the calculated strain rate using the original flow law parameters of Glen's flow law (Paterson, 1994) and the members of the modified flow law parameters of the composite flow law (Table 3). The calculated strain rates for Glen's flow law, dislocation creep and GBS-limited creep increase with increasing temperature and show a kink at their temperature threshold of 263K (Glen's flow law) or 262K (modified composite flow law). The strain rate increase

with temperature of the GBS-limited creep mechanism above the temperature threshold of 262K is considerably higher than for the dislocation creep mechanism or Glen's flow law, although this difference is largely related to our choice of Paterson's (1994) version of Glen's flow law. Using the Budd and Jacka (1989) description for the temperature sensitivity of Glen's law, would produce different results.

## 3.4 Calculated strain rates and deformation mechanisms

Figure 5 shows the calculated strain rates, along with the relevant microstructural data (Binder et al., 2013; Eichler et al., 2013; NEEM community members, 2013; Binder, 2014). The modified composite flow law shows that the calculated strain rate in the lower part of the glacial ice, which reaches to a depth of 2207 m (NEEM community members, 2013), is about $2.5 \cdot 10^{-11}$ s$^{-1}$ and the CPO has a strong single maximum. The dominant deformation mechanism of the modified composite flow law in the lower part of the glacial ice is GBS-limited creep, with only a very small contribution of dislocation creep to bulk strain rate. Glen's flow law predicts a higher strain rate (about $10^{-10}$ s$^{-1}$) than the modified composite flow law in the lower part of the glacial ice. At the interface between the glacial ice and Eemian-glacial facies, the calculated strain rate for the composite flow law drops by about an order of magnitude. The relative contribution of the two members of the composite flow law changes as well with GBS-limited creep and dislocation creep contributing roughly equally to bulk strain rate. At the same depth, the CPO changes from a strong single maximum in the glacial ice to a partial girdle in the upper part of the Eemian-glacial facies. The calculated strain rate of the modified composite flow law varies by about an order of magnitude between the finer and coarser grained regions close to the stratigraphic disruptions. This variation in calculated strain rate close to the stratigraphic disruptions is produced by GBS-limited creep, which is affected by the change in grain size. The strain rate produced by dislocation creep, which is not affected by the variation in grain size, steadily increases with depth throughout the Eemian-glacial facies. Glen's flow law, which is not affected by grain size variation either, predicts an increasing strain rate with depth and a higher strain rate than the modified composite flow law in the entire Eemian-glacial facies. The increase in strain rate coincides with the increase in temperature along the NEEM ice core (Figure 1 companion paper).

The relative contribution of GBS-limited creep and dislocation creep to the bulk strain rate of the modified composite flow law is roughly equal for the ice core sections that were assigned a mean grain diameter of 30 mm just below the stratigraphic disruptions at 2209.6 m and 2262.2 m of depth. At deeper levels, the contribution of GBS-limited creep to bulk strain rate for these coarse grained ice core sections increases. The increase in relative contribution of GBS-limited creep to bulk strain rate, at the assigned constant grain size of 30 mm, results from the lower activation energy at T>262K for dislocation creep compared to GBS-limited creep (Table 3, Figure 4). The difference in activation energy for Glen's flow law and the dislocation creep mechanism above their temperature thresholds is rather small (Table 3), which results in an almost similar order of magnitude strain rate increase with depth. The relative changes in strainrate between the different flow laws are very sensitive to activation energies used (Table 3). In the case of Glen's law and possibily for the dislocation creep mechanism, a higher activation energy could apply close to the melting temperature (e.g. Budd and Jacka 1989).

## 4 Discussion

The results show that the Eemian-glacial facies consists of layers with relatively fine grains that are alternated by layers of very coarse grains. The coarse grained layers have a strongly interlocking grain boundary structure with a partial girdle or multi maxima type of CPO, while the fine grained layers have a more regular grain shape and have a single maximum type of CPO (Figure 2 and 3). A comparison with other polar ice cores showed that layers with very coarse and interlocking grains with a multi maxima or partial girdle type of CPO start to appear at a T* value of about 262K (Table 1). The modified composite flow law of Goldsby and Kohlstedt (2001) predicts that the strain rate in the fine grained layers, which is almost entirely

produced by GBS-limited creep, is about an order of magnitude higher than the strain rate in the coarse grained layers, where GBS-limited creep and dislocation creep contribute roughly equally to bulk strain rate (Figure 5).

## 4.1 The role of impurities in premelted ice

It is well known that changes in impurity content correlate well with changes in mean grain size in polar ice cores (e.g. Fisher and Koerner, 1986; Paterson, 1991; Thorsteinsson et al., 1995; Cuffey et al., 2000); finer grains occur in ice with a higher impurity content. It is often assumed that impurities control grain size by pinning of grain boundaries (e.g. Fisher and Koerner, 1986; Gow et al., 1997; Durand et al., 2006), although the exact mechanism by which impurities control the mean grain size is still not understood in detail (Eichler et al., 2017). Similar to the Eemian-glacial facies in the NEEM ice core, the lower part of the GRIP ice core (Thorsteinsson et al., 1995) is likely affected by premelting (Table 1). In both the GRIP and NEEM ice core, the effect of impurities on grain size and shape is very large in the premelting regime (Thorsteinsson et al., 1995; NEEM community members, 2013). Due to the high impurity content in the glacial ice and late Eemian ice of the Eemian-glacial facies, the mean grain size in these layers remains relatively small. On the other hand, the low impurity Eemian ice has much larger grains. The second effect of the high impurity content in the finer grained layers in the Eemian-glacial facies is the melt content along grain boundaries and triple junctions is probably enhanced due to the lowering of the pressure-melting point by salts and/or impurities (e.g. Duval, 1977; Wettlaufer, 1999a, b; Döppenschmidt and Budd, 2000).

Impurities also provide additional interfaces, in addition to grain boundaries, where premelting can take place. These premelting films could act as dislocation sinks that may enhance dislocation motion in ice, preventing hardening by dislocation entanglement and thus enhance the strain rate. The type of impurity can change the effectivity of premelting, but as only few studies on particle species in solid polar ice are available so far (Ohno et al., 2005, 2006; Sakurai et al., 2009; 2011; Oyabu et al., 2015), this argument can only be made via the total surface of impurities per volume of ice.

## 4.2 The effect of premelting on ice microstructure

Premelting is expected to initiate at grain boundaries at temperatures below, but close to, the melting point (e.g. Orem and Adamson, 1969; Döppenschmidt et al., 1998). Since premelting itself, as well as the collection of water in veins (Nye and Mae, 1972), takes place at grain boundaries and triple junctions, grain boundaries are the major 'suspect' that enables weakening in polycrystalline ice. As temperature increases, a liquid-like amorphous layer gathers in layers along grain boundaries and veins along triple junctions, the presence of a liquid-like amorphous layer correlates with strain rates that are faster than predicted by extrapolation of low temperature results using the Arrhenius relation with a fixed activation energy (e.g. Mellor and Testa, 1969; Nye and Mae, 1972; Duval et al., 1977; Dash et al., 1995; De la Chapelle et al., 1999). Therefore, a flow law that is calibrated at lower temperature cannot simply be applied under premelting conditions, since the mechanical properties of the material at premelting temperature, apparently, are different. Due to the higher grain boundary surface area per unit of volume in a fine grained sample, the influence of premelting on a fine grained sample is expected to be stronger than for a coarser grained sample.

A small liquid-like amorphous layer at the grain boundaries may account for the increase of grain boundary mobility by about two to four orders of magnitude (De la Chapelle et al., 1998). Since grain boundary velocity is a function of grain boundary mobility and the driving force for grain boundary migration (e.g. Higgins, 1974; Alley et al., 1986), it is expected that a strong increase in grain boundary mobility leads to an increase in grain boundary velocity, provided that the stored strain energy for SIBM in the ice polycrystal is high enough. Ice deformation tests have indeed shown that grain boundary migration rates are high close to the melting point (e.g. Wilson and Zhang, 1996; Breton et al., 2016). Microstructural changes ocurring in ice close to the melting point that creates very coarse and interlocking grains, and a change in deformation behaviour, could both well be explained by the presence of a liquid-like amorphous layer that increases grain boundary mobility and consequently enhances SIBM. This enhanced SIBM would result in very coarse grains with an interlocking grain boundary

structure (e.g. Duval and Castelnau, 1995; Schulson and Duval, 2009; Breton et al., 2016), as observed in the layers with low impurity content of the Eemian-glacial facies (Figure 2b).

In addition to the effect of high temperature on grain boundary mobility, the cryostatic pressure in the lower part of polar ice cores (20-23 MPa for NEEM, Equation 6) will further enhance grain boundary mobility (Breton et al., 2016). Directed growth of migrating grain boundaries along subgrain boundaries in specimens deformed at a cryostatic pressure of 20 MPa can lead to a smaller median grain size and a more interlocking microstructure. The smaller grain size of samples deformed at a high cryostatic pressure of 20 MPa compared to samples deformed at atmospheric pressure could be caused by grain dissection (Breton et al., 2016; Steinbach et al., 2017), since this process also depends on SIBM.

The similarity between microstructures with coarse interlocking grains in the lower part of polar ice cores (Table 1, Figure 2b) and the ice microstructure developing during deformation tests close to the melting point suggests that these microstructures are governed by the same processes. It is therefore proposed that the sudden appearance of the coarse grains with an interlocking grain boundary structure in the lower part of polar ice cores is the result of premelting along the grain boundaries, which increases grain boundary mobility and consequently enhances SIBM.

### 4.3 Setting the temperature threshold for premelting

The in-situ temperature at which coarse and interlocking grains start to appear in polar ice sheets ($T^*$ about 262K) (Table 1), falls within the temperature range (258K to 263K) of the transition to a more temperature sensitive deformation mechanism during deformation tests (Mellor and Testa, 1969; Barnes et al., 1971; Weertman, 1983; Budd and Jacka, 1989; Paterson, 1994; Goldsby and Kohlstedt, 2001). The premelting temperature threshold for Glen's flow law (263K; Paterson, 1994) is 5K and 8K higher than the temperature thresholds for the dislocation creep (258K) mechanism and the GBS-limited creep mechanism (255K) applied by Goldsby and Kohlstedt (2001), respectively. Since the strain rate increase close to the melting point for dislocation creep and GBS-limited creep has been related to premelting along the grain boundaries (Goldsby and Kohlstedt, 2001), a similar temperature threshold can be expected for both deformation mechanisms.

The temperature threshold of 258K proposed by Goldsby and Kohlstedt (2001) for dislocation creep was taken from Kirby et al. (1987), who conducted ice deformation tests at a high confining pressure of 50 MPa. A confining pressure of 50 MPa lowers the pressure-melting point by 3.7K, using the pressure-melting constant for clean ice of $7.4 \ 10^{-8}$ K $Pa^{-1}$ (Hobbs, 1974; Weertman, 1983; Equation 5). This change in pressure-melting point due to the high confining pressure seems not to be considered by Kirby et al. (1987) and Goldsby and Kohlstedt (2001). Therefore, it is argued that the corresponding temperature threshold should have been 261.7K instead of 258K, if a confining pressure of 50 MPa is also assumed. This temperature threshold is much closer to 263K used in Glen's flow law (Paterson, 1994) and practically equal to 262K that was found by analysing ice microstructures of polar ice cores (Table 1). 261.7K is also closer to the temperature threshold proposed by Barnes et al. (1971), who found that the activation energy of ice is much higher between 271K and 265K (120 kJ mol[-1]) than between 265K and 259K (78.1 kJ mol[-1]). Similarly, 261.7K is also closer to Mellor and Testa (1969) and Budd and Jacka (1989) who found that above 263K the strain rate becomes progressively more temperature dependent when approaching the melting point.

### 4.4 Recrystallization and deformation mechanisms in the Eemian-glacial facies

The strain rate calculations using the composite flow law (Figure 5) show that the fine grained layers in the Eemian-glacial facies deform predominantly by GBS-limited creep. It is known that the basal slip system is not significantly affected by the cryostatic pressures that are reached in polar ice sheets (e.g. Rigsby, 1958; Cole, 1996). Therefore, basal slip is an important strain producing mechanism in both the coarse and the fine grained layers, but the way basal slip is rate limited is different. Since basal slip provides only two of five independent slip systems required for homogeneous deformation, at least one additional slip system or some other mechanism is required (e.g. Von Mises, 1928; Hutchinson, 1977). Grain boundary sliding

is favoured as the rate limiting mechanism for basal slip in the Eemian-glacial facies, since the grain boundary area in the impurity-rich and fine grained layers is relatively high and a higher grain boundary area enhances the effect of premelting on deformation behaviour (Barnes et al., 1971). However, compared to the glacial ice between 1419 m to 2207 m of depth (Figure 2b), the grain boundary network in the fine grained layers in the Eemian-glacial facies is relatively irregular with many bulges

(Figure 2a). The irregular grain boundary structure is likely a result of enhanced SIBM caused by premelting (Section 4.2). The occurrence of SIBM implies there are differences in internal strain energy between grains that are related to the basal slip component of deformation, which is still significant even when GBS is the rate controlling process. Still, the effect of impurities in these impurity-rich layers is strong enough to prevent the grains from growing into tens of millimetres, like the example of the coarse grained ice core section in Figure 2b, so grain boundary sliding remains an important rate limiting mechanism. Due

to the lower ratio of SIBM that is expected in these impurity-rich layers, the grains are longer-lived and can rotate towards a single maximum CPO (e.g. Van der Veen and Whillans, 1994). For these layers, the CPO is a reflection of cumulative strain, just like the CPO in the shallower parts of the ice sheet (e.g. Azuma and Higashi, 1985; Alley, 1988; Budd and Jacka, 1989; Llorens et al., 2016a, b, 2017).

For the coarse grained layers in the Eemian-glacial facies the calculations using the chosen creep laws predicts that

the deformation is not dominated by dislocation creep only, but that it has a substantial GBS-limited creep component going up to 75% at 2485 m depth (Figure 5). This would suggest that even in these coarse grained layers the creep behaviour is strongly grain size dependent. The question is whether this grain size dependence is in agreement with the interlocking grain boundary structure observed in the material (Figure 2b), since such a structure seems inconsistent with assumed sliding along grain boundaries. However, SIBM is highly active in these coarse grained layers and SIBM has been suggested to be a possible

accomodation mechanism for basal slip (Pimienta and Duval, 1987; De la Chapelle et al., 1999 ; Duval et al., 2000; Montagnat and Duval., 2000). If so, the flow law for GBS-limited creep is not applicable to these coarse grained materials. Alternatively, the original microstructure might have been obliterated by SIBM in situ. In that case, GBS might have locally rate limited basal slip (Raj and Ashby, 1971), hence basal slip might be rate limited by SIBM coupled by GBS (Drury et al., 1989), and the grain size dependent flow law ($\dot{\varepsilon}_{GBS}$) might still be of relevance.

The grain size and grain shape in these low impurity interglacial layers (Figure 2b) suggests that the grain boundaries are free to migrate at high SIBM rates (Duval and Castelnau, 1995; Schulson and Duval, 2009) which may be related to premelting (De la Chapelle et al. 1998) and are not or hardly influenced by impurities. The partial great circle to multi maxima CPO of these coarse grained ice core sections can be explained by the formation of new strain free grains with soft orientations (relatively high Schmid factor), which grow at the expense of grains oriented in a hard orientation (Alley, 1988; Durand et al.,

2009; Montagnat et al., 2015; Qi et al., 2017). This recrystallization mechanism can be described as discontinuous migration recrystallization (Wenk et al., 1989; Schulson and Duval, 2009) or SIBM-N (Faria et al., 2014a).  The dominant deformation mode deep in the ice sheet is expected to be simple shear, yet the CPO maxima in the coarse grained ice suggests that deformation in these layers may involve dominant co-axial strain rather than simple shear.

## 4.5 Strain localization in the Eemian-glacial facies

The ice in the Eemian-glacial facies alternates between fine grained ice with a single maximum CPO and coarse grained ice with a partial girdle CPO (Figure 5). The differences in grain size, grain shape and CPO in the Eemian-glacial facies implies that there are large differences in viscosity between the different layers. This is also suggested by the results using the modified composite flow law (Figure 5), that predict that the fine grained ice is much softer than the coarse grained ice when applying the modified composite flow law. Consequently, strain localization and strain partitioning into 'hard' and 'soft' layers can be

expected. Strain localization has been shown on many different scales in ice (e.g. Paterson, 1991; Wilson and Zhang, 1996; Grennerat et al., 2012; Jansen et al., 2016; Steinbach et al., 2016). The predicted strainrates in Figure (5) are based on flow laws for secondary creep at relatively low strain on samples with near random CPO. In addition to a variation in grain size, the

CPO has, depending on its orientation, a weakening effect on ice (e.g. Budd and Jacka, 1989; Faria et al., 2014b; Hudleston, 2015), so differences in CPO are likely to change the relative strain rates between the different ice layers shown in Figure 5. Deformation experiments with ice from other polar ice cores with a certain type of CPO, such as a single maximum or a multi maxima CPO, have different strain rates when the ice is deformed under different deformation modes. For example, simple shear deformation experiments on basal ice with a multi maxima CPO from Law Dome showed that the strain rates were comparable with those of isotropic ice (Russell-Head and Budd, 1979). The same study showed that samples from Law Dome with a single maximum CPO deformed much more readily than isotropic ice in simple shear (sheared in the direction of the measured surface velocity). The study of Lile (1978) with samples from Law Dome Summit and Cape Folger showed a similar result with samples containing a single maximum CPO that were deformed in simple shear (single maximum normal to the shear plane) showing much higher strain rates compared to laboratory-prepared isotropic ice with a similar grain size. For the case of domiant simple shear, which is expected to be the deformation mode in this part of the NEEM ice core (Dansgaard and Johnsen, 1969; Montagnat et al., 2014), different CPOs in the fine and coarse layers could lead to even higher strain rate differences, than predicted in Figure 5 from the Goldsby and Kohlstedt 2001 flow laws. The composite flow law that is used in this study does not explicitly include the effect of CPO on strain rate, while it is well known that the CPO has a weakening effect on ice depending on its orientation in certain different deformation modes (e.g. Budd and Jacka, 1989; Faria et al., 2014b; Hudleston, 2015).

The difference in microstructure between the impurity-rich ice and the low impurity ice poses an interesting hypothesis for the effect of premelting on ice dynamics close to the bedrock. In the case of the deeper ice where premelt may occur, from here on called the 'premelting zone', consists entirely of interglacial ice (with a low impurity content) the premelting zone is expected to be relatively viscous and hardly contributes to horizontal velocity, which will probably be localized above the premelting zone. In the case where glacial ice (with a high impurity content) is present in the premelting zone or the premelting zone consists entirely of glacial ice, a significant portion of the horizontal velocity will be accomplished in the premelting zone. The difference in microstructure of impurity-rich glacial and low impurity interglacial ice in the premelting zone can have important consequences for ice dynamics close to the bedrock. Since the impurity content in the Greenland ice sheet is approximately 10 times higher than the impurity content in the Antarctic ice sheet (e.g. Legrand and Mayewski, 1997), this effect of strain partitioning in the premelting zone is likely to be stronger in the Greenland ice sheet than in the Antarctic ice sheet.

The hypothesis of the different effects of impurity-rich and low impurity ice on ice dynamics in the premelting zone close to the bedrock is supported by borehole logging data from several ice cores. Data from Byrd station showed that the tilting rate (horizontal component of deformation rate) in the premelting zone (1810 m down to the bedrock at 2164 m of depth), where grains are coarse with a multi maxima CPO (Gow and Williamson, 1976), deformed much less than the remainder of the ice (Paterson, 1983). A similar observation for the EDML ice core was reported by Jansen et al. (2017). Morgan et al. (1997, 1998) also reported that the tilting rate at Law Dome decreased with depth as grain size increased and CPO deviated from a single maximum CPO. In this case the decreasing strain rate has been explained by a decrease of stress, caused by bed rock topography (Budd and Jacka, 1989). The highest simple shear strain rates over the entire depth profile in the Law dome (Dome Summit South) ice core, derived from the tilting, occur in a zone where the shear strain rates are otherwise decreasing. This layer is from the last glacial maximum and has a fine grain size and strong single maximum CPO. The zone of high shear strain rate cannot be explained by variations in temperature or stress, but it can be explained by the combined influence of CPO and grain size.

We speculate that strain partitioning between glacial and interglacial layers in the premelting zone could lead to a different rate of layer thinning, which could have important consequences for interpreting paleoclimatic records from polar ice cores. In the deepest part of the ice core the dating is based on ice flow modelling describing a homogeneus transition from

coaxial deformation to simple shear. It has often been shown that paleoclimatic records in the lower part of polar ice cores are disturbed (e.g. Alley et al., 1997; Gow et al., 1997; Suwa et al., 2006; Ruth et al.; 2007; NEEM community members, 2013), which could be caused by heterogeneous deformation and strain partitioning between glacial and interglacial layers as described above.

**5 Conclusions**

In many polar ice cores the grain size, grain shape and CPO change significantly in the ice close to the bedrock, which can have a large impact on the dominant deformation mechanism and the strain rate. Temperatures close to the bed rock are near the melting temperature and under these conditions enhanced creep and recrystallization can be explained by the occurrence of premelting along grain boundaries. In this study, actual temperature and grain size data of the NEEM ice core were used to
apply the composite flow law of Goldsby and Kohlstedt (2001) and the Paterson (1994) version of Glen's flow law to the lower part of the NEEM ice core with the aim of studying the effect of changes in grain size on the dominant deformation mechanism and the strain rate. Several stratigraphic disruptions are present in this part of the NEEM ice core, which causes layers of fine grained ice to be alternated by layers of coarse grained ice. After a microstructural evaluation of eight different polar ice cores from the Greenland and Antarctic ice sheets, it was found that microstructures that indicate premelting start at
a temperature (T*) of about 262K, which is within the temperature range at which a more temperature sensitive deformation mechanism starts to dominate the ice rheology in deformation tests. The composite flow law of Goldsby and Kohlstedt (2001) was modified to include this premelting temperature (T*) threshold of 262K.

The modified composite flow law predicts that, as a result of the grain size variation in the Eemian-glacial facies, the strain rate varies strongly with the fine grained layers deforming about an order of magnitude faster than the coarse grained
layers. In the fine grained glacial ice the dominant deformation mechanism is predicted to be GBS-limited creep. In the coarse grained interglacial ice, dislocation creep and GBS-limited creep contribute roughly equally to the bulk strain rate, with the contribution of GBS-limited creep to bulk strain rate increasing with depth. Glen's flow law, which is grain size insensitive, does not predict any variations in strain rate in the Eemian-glacial facies apart from a steadily increasing strain rate with depth caused by an increase in temperature with depth. Glen's flow law predicts a higher strain rate than the modified composite
flow law along the entire lowest 540 m of depth of the NEEM ice core. The magnitude of the strain rate increase would be stronger if a more realistic temperature dependence (e.g. Budd and Jacka 1989) for Glen's flow law is used.

Changes in grain size in the Eemian-glacial facies correlate strongly with changes in type and strength of CPO. The fine grained (<5 mm) impurity-rich glacial layers have a strong single maximum CPO, which is compatible with predominant simple shear deformation in this part of the NEEM ice core. The relatively fine grain size argues for GBS-limited creep to be
the dominant deformation mechanism in these layers. The coarse grained (>5 mm) low impurity layers have a partial great girdle to multi maxima type of CPO, which is more compatible with coaxial deformation than with simple shear deformation. Due to the coarse grains and the interlocking grain boundary structure, these layers are likely deforming by basal slip rate limited by recovery via SIBM, which removes dislocations and stress concentrations in grains and allows further deformation to occur, or by coupled SIBM and grain boundary sliding. Therefore, strain partitioning is expected in the Eemian-glacial
facies with the fine grained layers with a single maximum CPO deforming at high strain rates, while the coarse grained interglacial layers with a partial girdle type of CPO deform at much lower strain rates. The difference in microstructure, and consequently difference in viscosity, of impurity-rich and low impurity ice in the premelting zone can have important consequences for ice dynamics close to the bedrock.

**Acknowledgements**

This work has been carried out as part of the Helmholtz Junior Research group "The effect of deformation mechanisms for ice sheet dynamics" (VH-NG-802). The NEEM light microscope data used in this study has been made available by www.pangaea.de. The authors would like to thank Sepp Kipfstuhl and Tobias Binder for providing data and encouraging discussions. David Prior and Adam Treverrow are thanked for elaborate and very helpful comments on the first versions of the manuscript. The authors would like to thank all the NEEM Community members who were involved in the preparation of the physical properties samples in the field. This work is a contribution to the NEEM ice core project which is directed and organized by the Center of Ice and Climate at the Niels Bohr Institute and US NSF, Office of Polar Programs. It is supported by funding agencies and institutions in Belgium (FNRS-CFB and FWO), Canada (NRCan/GSC), China (CAS), Denmark (FIST), France (IPEV, CNRS/INSU, CEA and ANR), Germany (AWI), Iceland (RannIs), Japan (NIPR), South Korea (KOPRI), the Netherlands (NWO/ALW), Sweden (VR), Switzerland (SNF), the Unites Kingdom (NERC) and the USA (US NSF, Office of Polar Programs).

**Code/Data availability**

For the model described in this paper we used Excel, with the NEEM grain size and temperature data from Pangaea Data Publisher for Earth & Environmental Science (https://doi.org/10.1594/PANGAEA.838063, https://doi.pangaea.de/10.1594/PANGAEA.743296) and with the flow law parameters given in table 3.

**Author contribution**

EJK prepared the code, the data, the set-up of simulations and conducted model runs. JHPdB developed the algorithm for the code and provided the initial idea. MRD and GMP supervised and initiated the simulation set-ups and interpretations. JE and IW provided ice core samples and data, glaciological background and data preparation. All authors jointly interpreted results and wrote the manuscript.

**Competing interests**

No competing interests.

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

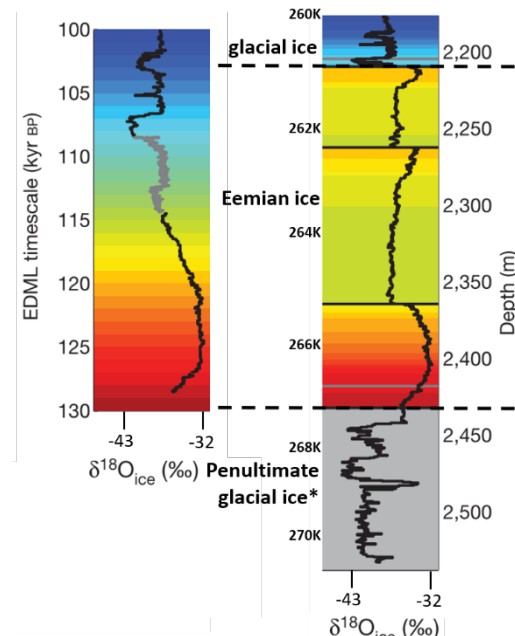

**Figure 1:  On the left the reconstructed water stable isotope ($\delta^{18}O_{ice}$) record from NEEM on the EDML1 timescale. The grey curve between 114-108 kyr BP is NGRIP data. The background has been color-coded based on age.  On the right the NEEM stable isotope ($\delta^{18}O_{ice}$) record on the original depth scale. No dating was attempted below 2432.2 m of depth. The two dotted lines separate the glacial, Eemian and what is likely the Penultimate glacial ice (indicated by \*). The in-situ temperature at several depths in the NEEM ice core is indicated. Figure based on NEEM community members (2014).**

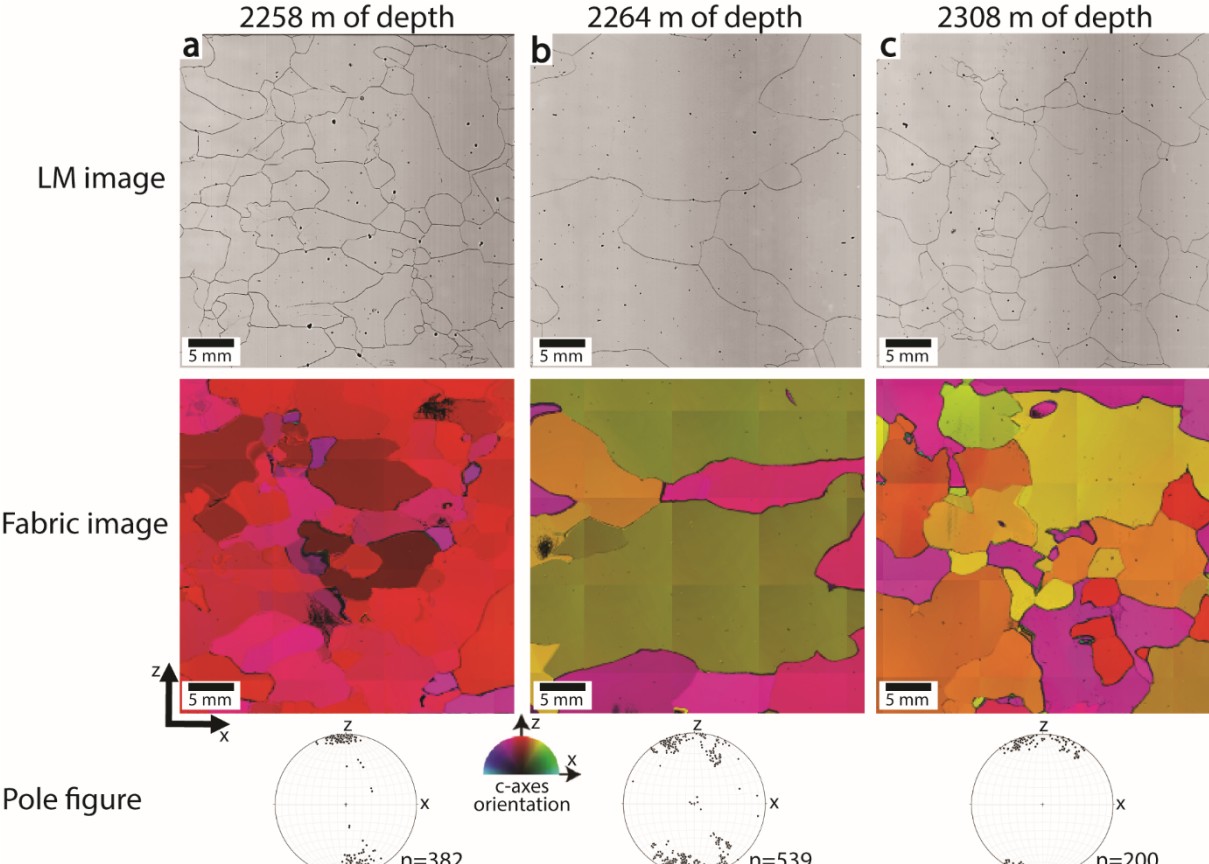

**Figure 2: The LASM, FA images and pole figures from three different ice core sections in the Eemian-glacial facies. The ice core sections in a) and b) are just above (2258 m of depth) and below (2264 m of depth) the stratigraphic disruption at 2262.2 m of depth, while c) is from the middle of one of the overturned layers (2308 m of depth). The z-axis is parallel to the (vertical) ice core axis. The number of grains n included in the whole (90 x 55 mm) orientation image and pole figure is provided next to the pole figure (equal area, vertical plane, one point per grain). The pole figure in (b) contains the c-axes data from six consecutive ice core sections to increase the number of grains in the pole figure. LM images taken from Kipfstuhl (2010) (doi:10.1594/PANGAEA.743296). Fabric images and pole figures taken from Weikusat et al. (2010) (doi:10.1594/PANGAEA.744004).**

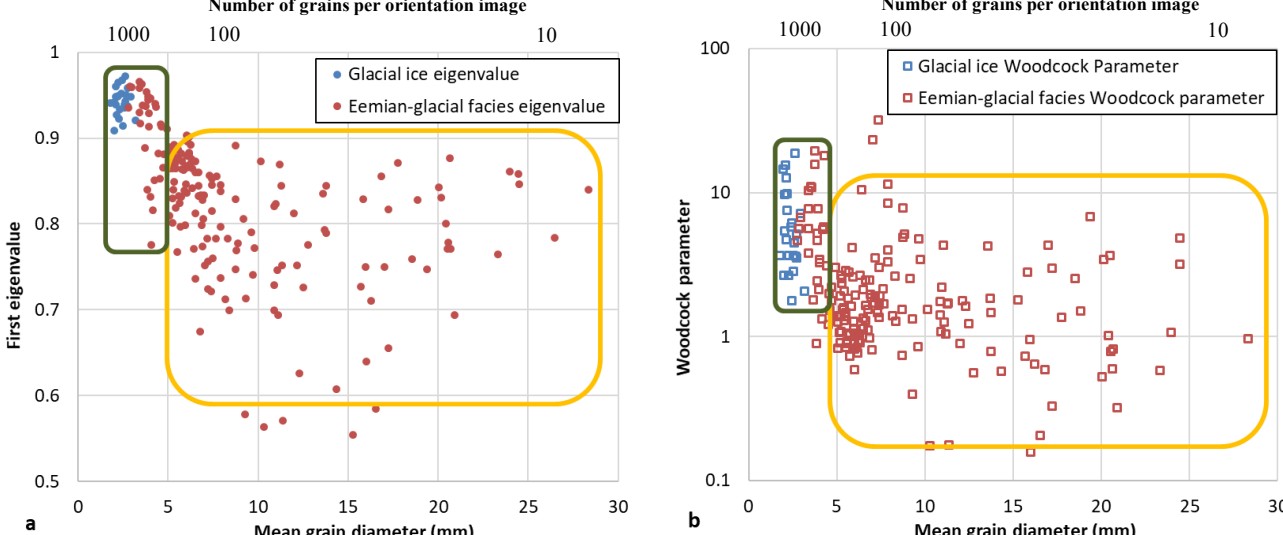

**Figure 3: (a) The first c-axis eigenvalue (dots) and (b) the Woodcock parameter (open squares) per FA image versus the mean grain diameter of the ice core sections in the lower part of the glacial ice (2000-2207 m of depth) in blue and the Eemian-glacial facies (2207-2540 m of depth) in red. The number of grains per orientation image was estimated by dividing the area of the orientation image by the mean grain area. Two classes of grains can be distinguished based on mean grain diameter, eigenvalue and Woodcock parameter. One class (green rectangle) has a fine mean grain size <5 mm and a first eigenvalue and Woodcock parameter that is comparable to the glacial ice and the other class (yellow rectangle) with a larger mean grain diameter (>5 mm) that has a lower eigenvalue and predominantly lower Woodcock parameter.**

**Table 1: Polar ice cores drilled at the Greenland and Antarctic ice sheets that show a sudden increase in grain size and change in CPO in the lower part of the ice core. The depth where these changes occur is given, together with ice core length, bottom borehole temperature, the in-situ temperature and the pressure-corrected temperature (T\*) at the sudden grain size increase and change in CPO and the age of the ice at the transition to a sudden grain size increase and change in CPO. The age of the ice is given in marine isotope stages (MIS) and approximate ka BP (Lisiecki and Raymo, 2005).**

| Name of ice core | Ice core length (m) | Borehole bottom T (°C) | Depth grain size increase and/or CPO change (m) | In-situ T at grain size increase and/or CPO change (°C) | Pressure-corrected temperature T* (°C) | Age of ice at transition to enhanced SIBM microstructure |
|---|---|---|---|---|---|---|
| Byrd[1] | 2164 | -1.6* | 1810 | -13 | -11 | MIS 3 (29-57 ka) |
| EDC[2] | 3270 | -2.3* | 2812 | -13 | -11 | MIS 12 (424-478 ka) |
| EDML[3] | 2774 | -3* | 2370 | -13 | -11 | MIS 6–MIS 5e (123-130 ka) |
| GISP2[4] | 3053 | -9 | 2950 | -14 | -11 | MIS 6-MIS 5 (~130 ka) |
| GRIP[5] | 3029 | -9 | 2790 | -13 | -11 | MIS 5e-MIS 5d (109-123 ka) |
| NEEM[6] | 2540 | -3.4 | 2207 | -12 | -10 | MIS 5e-MIS 5d (109-123 ka) |
| Siple dome[7] | 1004 | -1.3 | 605 | -13 | -12 | MIS 1 (~14 ka) |
| WAIS[8] | 3405 | unknown | 3000 | just below -10** | not applicable | MIS 3 (29-57 ka) |

**\*Subglacial water encountered during drilling close to the bedrock.**
**\*\*A specific temperature was not given, only that the grain size increase started just below -10°C.**
**[1]Gow and Williamson (1976); Hammer et al. (1994); Gow and Engelhardt (2000); Epstein et al. (2011).**
**[2]EPICA community members (2004); Augustin et al. (2007); Durand et al. (2009).**
**[3]Ruth et al. (2007); Weikusat et al. (2017).**
**[4]Gow et al. (1997); Suwa et al. (2006).**
**[5]Johnsen et al. (1995); Thorsteinsson et al. (1997); Suwa et al. (2006).**
**[6]NEEM community members (2013); Sheldon et al. (2014).**
**[7]Nereson et al. (1996); Gow and Engelhardt (2000).**
**[8]Fitzpatrick et al. (2014); Buizert et al. (2015).**

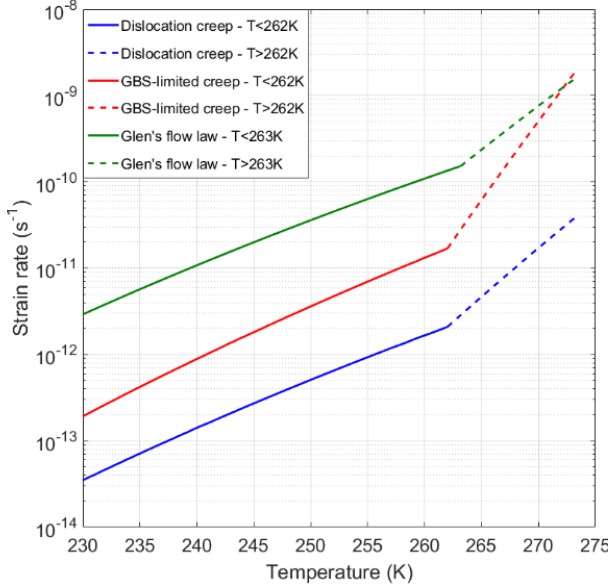

**Figure 4: Log strain rate versus temperature for Glen's flow law and the two mechanisms (dislocation creep and GBS-limited creep) that form the members of the modified composite flow law, using the flow law parameters from Table 3. A stress of 0.07 MPa and a mean grain diameter of 5 mm were used to calculate the strain rate.**

**Table 2: The flow law parameters of Glen's flow law (Paterson, 1994) and the end members of the original composite flow law for dislocation creep and GBS-limited creep. The corrected flow law parameters used in the companion paper are also shown below the dotted line. The flow law parameters used in this paper are shown in table 3.**

| Creep regime | A (units) | n | p | Q (kJ mol$^{-1}$) | 10 |
|---|---|---|---|---|---|
| Glen's flow law (T<263K) | $3.61\ 10^{5} MPa^{-3.0}s^{-1}$ | 3.0 | 0 | 60 | |
| Glen's flow law (T>263K) | $1.73\ 10^{21} MPa^{-3.0}s^{-1}$ | 3.0 | 0 | 139 | |
| Dislocation creep (G&K01) T<258K | $1.2\ 10^{6} MPa^{-4.0}s^{-1}*$ | 4.0 | 0 | 60 | |
| Dislocation creep (G&K01) T>258K | $6.0\ 10^{28} MPa^{-4.0}s^{-1}$ | 4.0 | 0 | 181 | 15 |
| GBS-limited creep (G&K01) T<255K | $3.9\ 10^{-3} MPa^{-1.8}m^{1.4}s^{-1}$ | 1.8 | 1.4 | 49 | |
| GBS-limited creep (G&K01) T>255K | $3.0\ 10^{26} MPa^{-1.8}m^{1.4}s^{-1}$ | 1.8 | 1.4 | 192 | |
| Dislocation creep (comp. paper) | $5.0\ 10^{5} MPa^{-4.0}s^{-1}$ | 4.0 | 0 | 64 | |
| GBS-limited creep (comp. paper) | $3.9\ 10^{-3} MPa^{-1.8}m^{1.4}s^{-1}$ | 1.8 | 1.4 | 49 | 20 |

**\*This value of A was updated from the original value (Goldsby, 2006).**

**Table 3: The flow law parameters of Glen's flow law (Paterson, 1994) and the members of the modified composite flow law for dislocation creep and GBS-limited creep of which some parameters remained the same as reported in Goldsby and Kohlstedt (1997, 2001). For this paper, the flow law parameters were contrained with the temperature threshold of 262K, which results in a change in activation energy and the pre-exponential factor for dislocation creep compared to the flow law used in our companion paper.**

| Creep regime | A (units) | n | p | Q (kJ mol$^{-1}$) |
|---|---|---|---|---|
| Glen's flow law (T<263K) | $3.61 \ 10^5 MPa^{-3.0}s^{-1}$ | 3.0 | 0 | 60 |
| Glen's flow law (T>263K) | $1.73 \ 10^{21} MPa^{-3.0}s^{-1}$ | 3.0 | 0 | 139 |
| Dislocation creep (T<262K) | $5.0 \ 10^5 MPa^{-4.0}s^{-1}$ | 4.0 | 0 | 64 |
| Dislocation creep (T>262K) | $6.96 \ 10^{23} MPa^{-4.0}s^{-1}$ | 4.0 | 0 | 155 |
| GBS-limited creep (T<262K) | $1.1 \ 10^2 MPa^{-1.8}m^{1.4}s^{-1}$ | 1.8 | 1.4 | 70 |
| GBS-limited creep (T>262K) | $8.5 \ 10^{37} MPa^{-1.8}m^{1.4}s^{-1}$ | 1.8 | 1.4 | 250 |

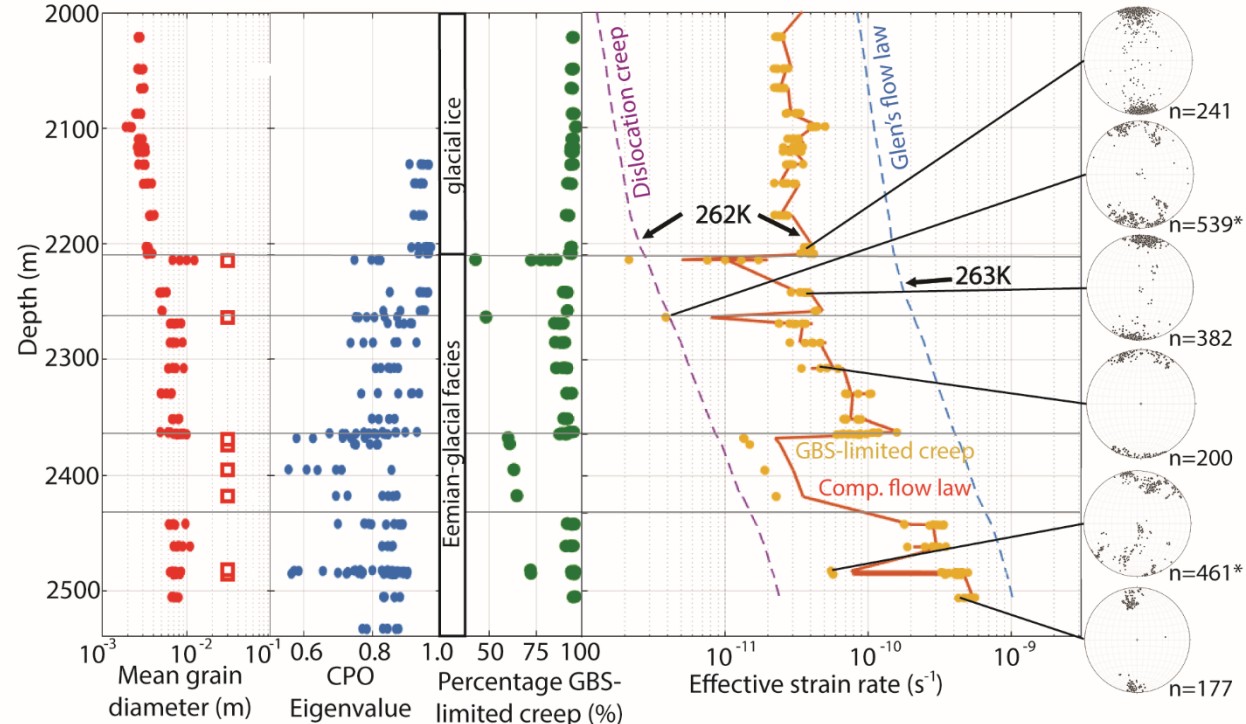

**Figure 5: Results showing as a function of depth for the lowest 540 m of the NEEM ice core, the mean grain diameter (red dots), first CPO eigenvalues (blue dots), calculated strain rates (Table 3) using Glen's flow law (blue dotted line), the modified composite flow law (orange line) and the two deformation mechanisms of the Goldsby and Kohlstedt flow law, dislocation creep (purple dotted line) and GBS-limited creep (yellow dots), (where, the orange line shows the total strain rates made up of the dislocation creep strain rate, shown by the purple line and the GBS-limited strain rate, shown by the yellow dots). The percentage contribution of GBS-limited creep to bulk strain rate is also shown (green dots). A red open square instead of a red dots was used when the grains size was too large to determine a mean grain diameter by the Ice-image software and a mean grain diameter of 30 mm was used. The four horizontal grey lines represent the stratigraphic disruptions that were found in the NEEM deep ice core (NEEM community members, 2013). The glacial ice extends to a depth of 2207 m, with the glacial-Eemian facies continuing until the ice-bedrock interface at 2540 m depth. The depth of the temperature (T\*) threshold for the modified composite flow law (262K) and Glen's flow law (263K) are indicated. On the right, six pole figures (equal area, vertical plane, one point per grain) at different depths are shown with the number of grains (*n*) next to the pole figure. At two depths, indicated by an asterisk (\*), the CPO data from six consecutive ice core sections were combined to increase the number of grains in the pole figure. Pole figures in last panel taken from Weikusat et al. (2010) (doi:10.1594/PANGAEA.744004).**