# Peer review of "Using a composite flow law to model deformation in the NEEM deep ice core, Greenland: Part 2 the role of grain size and premelting on ice deformation at high homologous temperature"

_The Cryosphere, 2018_

## Referee Comment (RC1) · David Prior (Referee) · 17 Apr 2019

Review of: **Kuiper**, de Bresser, Drury, Eichler, Pennock and Weikusat: "Using a composite flow law to model deformation in the NEEM deep ice core, Greenland: Part 2 the role of grain size and premelting on ice deformation at high homologous temperature"

By **Dave Prior** University of Otago.

This is an excellent paper. The quantitative analysis of potential balance of grain size sensitive and grain size insensitive deformation mechanisms in different grain size layers in the deeper, warmer section of the NEEM ice core is very important. The analysis shows that finer grained layers should deform more rapidly and with a greater proportion of deformation attributed to the grain size sensitive mechanisms. This is a crucial insight, as this deep core section may be representative of the warm basal regions of glaciers and ice sheets where strain is maximized/ deformation localized; such sections may dominate the rheology of glaciers and ice sheets. The paper does need some substantial re-writing, the area where the paper needs most modification is in discussion around premelt. Overall the paper is rather rambling and overlong and would benefit from significant shortening and tightening.

I have also reviewed the part one paper and I think the authors decision to separate the two papers is a good one. The outcomes are clearer and impacts are more effective as two papers. There are a number of comments related to the Part 1 paper that are also applicable to this Part 2 paper. I have copied these to the end of the review.

I also have an annotated pdf the authors can have.

**Premelting**

The writing about premelting needs some significant modification. The key problem is that what you are describing, both in your own data and from the literature, is a change in the kinematics of processes as a function of temperature. These are observations. Attribution of these obeservations to premelt is an interpretation and much of your writing does not make this important distinction. I believe the premelt interpretation, but it is important that we make it clear that there is little direct evidence for premelt in ice. The best (almost) direct evidence I know for premelt in grain boundaries is the Raman spectroscopy data presented by (Hammonds and Baker, 2018) that shows an aqueous phase on triple junctions in sulphuric acid doped ice. The review by Dash et al (in your reference list) is extremely thorough. The optical measurements of liquid film thickness on the basal plane surface provide some direct evidence but most of the paper outlines reasonable physical inference of premelting. A good example of misleading writing in your paper is lines 27 to 29 on page 2: *"High temperature deformation tests on polycrystalline ice have shown that a small liquid-like amorphous layer at the grain boundary increases grain boundary mobility by two to four orders of magnitudes (Duval and Castelnau, 1995; Schulson and Duval, 2009)"*. Castelnau and Duval do not mention

premelt or any related concept in their paper. They do discuss changes in recrystallisation behaviour at ~ -10C but they do not talk about "liquid -like" or "amorphous" layers. I have had a re-scan through the Schulson and Duval book and this is not a suitable reference for premelting. As far as I can tell the only explicit mention of premelting is on page 231 and is in the context of crack propagation. There is a brief discussion of the effect of a liquid phase on secondary creep on page 127, but this appears to be talking about ice-melt mix, rather than grain boundary pre-melt. The sections on recrystallisation and particularly on GBM (pages 130-138) make no mention of pre-melt or grain boundary properties. I have not had time to check each and every reference you cite. It's really important that the writing is clear and citations are used correctly. The writing should distinguish:

1. Direct evidence of premelt in the paper cited.
2. Indirect evidence in the paper cited, that the authors of that paper interpret in terms of premelt (e.g. Vaughan et al., 2016 ultrasonsonic attenuation see below).
3. Phenomena presented in a paper that you can reasonably interpret in a premelt framework. The Duval and Castelnau, 1995 paper would fall into this category.

There is an area of research that can be interpreted in terms of premelt that you do not mention. This is the work on attenuation of sound waves, where the attenuation is much more effective at warmer temperatures. Key references include (Peters et al., 2012), (Kuroiwa, 1964) and (Vaughan et al., 2016). In fact the Vaughan et al paper has been analysed in terms of the changing contribution to the bulk stiffness tensor of grain boundary elasticity by (Sayers, 2018): this can be interpreted as a parameterization of premelt related grain boundary properties. There is good discussion relevant to this topic in (McCarthy and Cooper, 2016).

An interesting additional discussion point is on whether premelting is a threshold. The splitting of activation energy into a high and low temperature values is rather artificial- it's based on limitations of the experimental data sets we have. I have a strong feeling that activation energy changes continuously from some low T value at ~-20C and lower through progressively higher values as T increases (see also discussion in the Cuffey and Paterson text book). If this is true and the T dependence is a proxy for grain boundary properties related to premelt then it suggests that pre-melt is not a simple threshold phenomenon.

**A schematic overview at start**

The paper needs an introductory schematic overview figure of the microstructures and grain sizes in the NEEM core with emphasis on the lower section. This can be used to highlight the structural and stratigraphic complexities, making the text easier to follow and can be used to put the layers with different grain size characteristics in context. Such a figure will increase the impact and uptake of the paper significantly.

**Grain numbers in fig 1.**

I am confused by the numbers of grains in the seteronets in fig 1, compared to the number I see in either the LM or the fabric image. For (c) I count about 40 grains, whereas the stereonet contains 200. For (a) I count about 70 grains, whereas the stereonet contains 382. For (b) I count ~14 for the single frame shown whereas the

stereonet for six frames has 539 grains giving an average of ~90 grains per frame. Please explain these in the manuscript.

**Woodcock parameter**

You need to explain this a little more completely. It is not explicit from Nigel Woodcock's original paper what this parameter is (of course he does not name it that way). You need an equation that explains how this parameter relates to the principal eigenvalues.

**Names for the modified flow laws.**

This is repeated later in the cut from Part 1 – but I want to expand a little on it here.

I think you need short names that clearly distinguish the different flow law fits. This becomes particularly important when one considers the two parts of your work as the second paper has a different fit (for justifiable reasons). Something like

- G&K: original Goldsby and Kohlstedt flow laws.
- $G\&K_{corr}$: Goldsby and Kohlstedt flow laws corrected as in part 1.
- $G\&K_{262}$: Goldsby and Kohlstedt flow laws with best fit for 262K switch (related to the part 2 paper)

So in this Part 2 paper I think you need to list the original G&K parameters as well as your modified parameters.

**Eemian Glacial Facies vs Eemian ice.**

I got rather confused in the distinction or not of these terms. Please attempt to make this as clear and easy for someone to understand as possible. Also it is important that the reader can recheck their understanding of these terms quickly- i.e. not needing to read a length of text.  The schematic figure I suggest would be a good place to make the meaning of these terms clear. Then there is a fast reference for the reader to go back to.

**Strain energy during GBS**

All of the micrographs show grains with irregular grain boundaries, suggesting that GBM is operative. In this context I think you need a short discussion on how you generate the internal strain energy to drive this under conditions where GBS contributes >90% of the strain rate.

***Split Fig 2 into 2 graphs***: one for first eigenvalue one for Woodcock parameter. Or abandon one of the measures.

***Fig 4.*** Get rid of the box legend. Red, blue and green dots are already labelled by the horizontal axes. Dislocation Creep, GBS and Glen would be better labeled by words

written along the trajectory of these lines. This is also a good example of a caption that is way too long.

**Minor things: through manuscript**

Page 2 line 1-2. CPO development and weakening are correlated in experiments, but there is no clear evidence that the CPO formation causes the weakening. Please be careful with the way you say this.

Page 6 line 4. The explanation here quotes units with joules. The actual units you use have the energy component in kJ (table). It would be simpler to keep the units the same.

Page 7 line 30. I think you would be better saying two classes of microstructure?

Page 8 line 30. Glen's T dependency is fairly crude so I don't think you can quote him for a change in Q at 263K. Cuffey & Paterson? Where you quote this kink please explain briefly then basis for it, rather than just quoting the reference. In Table 2 Q < 263 is quoted as 60kJmol-1. I thing originally this is from Paterson 1977, summary of field data, not from the text book.

Page 14, line 14. It is stated in various reviews that CPO controls weakening in different orientations. There is actually very little data that demonstrates this. Most that is commonly quoted either does not explore different orientations or has other changes (e.g. grain size) that can also contribute to weakening. For me the paper that gets closest do demonstrating the orientation effect is (Azuma, 1995).

**Comments copied and pasted from the Part 1 review: that are also relevant here.**

**"Accommodated" by**

Expressions such as "grain boundary sliding accommodated by easy slip" are commonly used by the rock deformation community. The problem is that this terminology is not used consistently. I find this language highly uninformative. If it is used to indicate a mechanism dependency then which is the "dependent" mechanisms depends on how you understand the English: different readers interpret it in opposite ways. Furthermore some use this terminology to indicate the mechanism within the grain boundary (as opposed to a kinematically required partner mechanism) as discussed in some of the original GBS literature from Michael Ashby (see for example fig 6.1 in Schulson and Duval, 2009). In your paper the language becomes particularly confusing through variation of language used - especially bearing in mind that many of the readers are not from the rock deformation community. This language discussion arises repeatedly and I recall a meeting back in 2006 where I was involved in extensive discussions with at least for the two co-authors on this topic. There are several statements that inform what language might be useful:

- Grain boundary sliding of a polycrystalline material (where pore spaces are not allowed) requires that the individual crystals change shape.
- Where a polycrystalline deforms by a mechanism that restrict the shape change of each individual crystal (e.g. glide on one crystal plane and homogenous bulk strain), grain boundary sliding is required.
- Diffusion creep in a polycrystalline material requires grain boundary sliding.

You are primarily trying to explain the flow law form:

$$\left(\frac{1}{\dot{\varepsilon}_{basal}} + \frac{1}{\dot{\varepsilon}_{gbs}}\right)^{-1}$$

embedded within equation (2).  The explanation on lines 6 and 7 of page 5 are not going to help the reader understand this. The way I usually explain this mechanism is that GBS is accompanied by basal slip. The two mechanisms are dependent upon each other, one cannot proceed without the other. The explanation on line 7 is particularly confusing as it indicates (wrongly) that both of the inverse terms inside the brackets each involves both basal slip and GBS.

$\frac{1}{\dot{\varepsilon}_{basal}}$ is just the inverse of the strain rate related to basal slip. GBS is not involved.

$\frac{1}{\dot{\varepsilon}_{gbs}}$ is just the inverse of the strain rate related to GBS. Basal slip is not involved.

It is the expression as a whole that provides the rate dependence. So that if

- $\dot{\varepsilon}_{basal} \gg \dot{\varepsilon}_{gbs}$ then $\left(\frac{1}{\dot{\varepsilon}_{basal}} + \frac{1}{\dot{\varepsilon}_{gbs}}\right)^{-1} \approx \dot{\varepsilon}_{gbs}$ ie GBS limits the strain rate

- $\dot{\varepsilon}_{basal} \ll \dot{\varepsilon}_{gbs}$ then $\left(\frac{1}{\dot{\varepsilon}_{basal}} + \frac{1}{\dot{\varepsilon}_{gbs}}\right)^{-1} \approx \dot{\varepsilon}_{basal}$ ie basal slip limits the strain rate.

You use the "rate limiting" terminology (in addition to the accommodation terminology) and this language is much more satisfactory to me. I think that you can make the paper much clearer by abandoning the "accommodated by" expression and describing the mechanism balance in terms of rate limits.

**The "Glen" law**
I think you need to take care with the language used related to the Glen law. Citing Glen (1955) for a Glen law with n=3 does a disservice to John Glen. Glen's three key papers have n values of 4 (1952), 3.3 changing to 4 (1953) and 3.2 to 4.2 (1955) respectively. As far as I know Glen has not written that one should use an n=3 relationship; if anything, he suggests that n values for naturally deforming ice should be around 4. So, the n=3 is a simplification

of Glen's work that is in common use (I'm not really sure who did this first). The Glen law in common use has n=3 but it was not Glen who set this value. It would be nice if your introduction of the Glen law made this subtlety clear.

**Girdle**

When you use the term girdle to describe a CPO element can you describe this more completely. Girdle covers a wide range of things on a stereonet. I restrict the term for great circle distributions, but many include small circle distributions under this name. Even if more restricted some information on "girdle" orientation would be useful.

**CPOs during GBS in ice.**

The discussion ocould make reference to a paper by one of my students.(Craw et al., 2018) show incredibly strong CPOs develop at relatively low strain (20% shortening) in large grains. In this case the large grains are not strongly strained (they do not have elongated shapes) and the large grains are surrounded by a network of fine recrystallized grains that have an equivalent but much weaker CPO. In that paper we suggest that GBS is an important mechanism controlling the microstructural evolution but some slip on the basal plane of the large grains is needed to develop such a strong CPO.

**Figure Captions**

Generally figure captions are way too long and include discussion elements that should be in the main text. The role of the figure caption should be to explain what is in the figure, where that is not clear from the figure itself. Discussion of the significance of a figure should be in the text.

Azuma, N., 1995, A flow law for anisotropic polycrystalline ice under uniaxial compressive deformation: Cold Regions Science and Technology, v. 23, no. 2, p. 137-147.

Craw, L., Qi, C., Prior, D. J., Goldsby, D. L., and Kim, D., 2018, Mechanics and microstructure of deformed natural anisotropic ice: Journal of Structural Geology, v. 115, p. 152-166.

Hammonds, K., and Baker, I., 2018, The Effects of H2SO4 on the Mechanical Behavior and Microstructural Evolution of Polycrystalline Ice: Journal of Geophysical Research-Earth Surface, v. 123, no. 3, p. 535-556.

Kuroiwa, D., 1964, Internal friction of ice: Contrib. Inst. Low Temp. Sci. Hokkaido Univ., Ser. A,, v. 18, p. 1-24.

McCarthy, C., and Cooper, R. F., 2016, Tidal dissipation in creeping ice and the thermal evolution of Europa: Earth and Planetary Science Letters, v. 443, p. 185-194.

Peters, L. E., Anandakrishnan, S., Alley, R. B., and Voigt, D. E., 2012, Seismic attenuation in glacial ice: A proxy for englacial temperature: Journal of Geophysical Research-Earth Surface, v. 117.

Sayers, C. M., 2018, Increasing contribution of grain boundary compliance to polycrystalline ice elasticity as temperature increases: Journal of Glaciology, v. 64, no. 246, p. 669-674.

Vaughan, M. J., van Wijk, K., Prior, D. J., and Bowman, M. H., 2016, Monitoring the temperature-dependent elastic and anelastic properties in isotropic polycrystalline ice using resonant ultrasound spectroscopy: Cryosphere, v. 10, no. 6, p. 2821-2829.

---

## Referee Comment (RC2) · Adam Treverrow (Referee) · 28 Apr 2019

Manuscript review: tc-2018-275

**Using a composite flow law to model deformation in the NEEM deep ice core, Greenland: Part 2 the role of grain size and premelting on ice deformation at high homologous temperature**

**Ernst-Jan N. Kuiper, Johannes H. P. de Bresser, Martyn R. Drury, Jan Eichler, Gill M. Pennock, Ilka Weikusat**

**Reviewer: Adam Treverrow**

**General comments**

The manuscript describes the use of two flow relations to calculate deformation rates at depths from 2000–∼ 2500 m in the NEEM ice core. The novel aspect of this manuscript is the application of the Goldsby and Kohlstedt (2001) flow relation, using ice microstructure and temperature data to investigate variablity in deformation mechanisms and strain rates over small spatial scales – and the factors that may contribute to this. In general this is a great paper which will be a valuable contribution to the the Cryosphere and I enjoyed reading it. This work should motivate further discussion and research into the effects of microstructure and temperature on ice deformation, and the extent to which localisation plays a role in large-scale ice sheet dynamics.

The paper is well written and thorough (good background and referencing), nicely presented and has few errors (grammar, syntax). The quality of the writing does decrease in sections 4.4 and 4.5 of the Discussion and the Conclusions. Primarily, these sections could be more concise, and more importantly, attention should be paid to being more explicit about which are the speculative aspects of the interpretation and discussion - at times speculative comments are presented in such a way that the casual reader may interpret these statements as observations or facts. I don't think this is deliberate, but the tone of the writing should to be consistent. Some specific examples of the issues noted above are given in the comments below.

Finally, where appropriate, the limitations of the study design need to be clearly addressed; these are also dealt with in the following specific comments. The paper is largely in good shape, so many of the following comments relate only to minor issues. However, these and the several more significant issues should be addressed prior to publication.

**Specific comments**

In the comments that follow text, where copied from the manuscript, is emphasised using italics.

   **P1 L2, L3**: In the Abstract there are references to layers of *fine grained glacial ice* alternated with *coarser grained Eemian ice*. While these designations are described in the main text, the fact that they're not defined in the abstract will create uncertainty for readers not familiar with the use of climatic periods (e.g. *glacial* and *Eemian*) to describe types of ice, or those that don't immediately recall how the periods relate to one another. The abstract should be revised to improve clarity, e.g. *ice deposited during the last glacial period* (as used elsewhere in the text) would be an improvement. While I understand why the terms *glacial*, *Eemian* and *Eemian-glacial facies* were used throughout the manuscript, an alternative terminology based directly on impurity content or microstructure could be used and should at least be considered.

   **P1 L15-18**: *The dominant deformation mechanisms between the layers is also different with basal slip accommodated by grain boundary sliding (GBS-limited creep) being the dominant deformation mechanism in the glacial layers, while GBS-limited creep and dislocation creep (basal slip accommodated by non-basal slip) contribute both roughly equally to bulk strain in the coarse grained layers.*

Do you really know this from direct observation or is this simply the result you get when you use the flow relation of Goldsby and Kohlstedt (2001) in conjunction with the microstructure observations? This sentence needs to be rewritten to make it clear this is the 'predicted' balance between deformation mechanism when you use the Goldsby and Kohlstedt (2001) flow relation and the NEEM microstructure data.

**P1 L19**: *...impurity-depleted...* is ambiguous since it also suggests that some depletion process that reduces the impurity concentration might have occurred. Since the impurity content of the Eemian ice is simply low relative to the *glacial ice*, it would be better to use 'low(er) impurity Eemian ice', or similar, throughout the manuscript.

**P2 L1**: The use of *...shallower ice...* is ambiguous since it is also suggestive of ice that is thinner, not just ice that is closer to the surface, which is what I think this sentence is about. As it stands there's also some (small) potential for things to get mixed up in the concept of the shallow ice approximation (SIA).

**P2 L2**: Shear stress does increase with depth, but not all the way to bedrock. If it did then simple shear strain rates would increase down to the ice-bedrock interface, but strain rates derived from borehole inclination measurements show that they don't....which is the point of the following sentence on P2 L3. Some revisions is required here for consistency with the sentence commencing at P2 L3. Also *bedrock variations* is somewhat vague. Is this a reference to bedrock topography, rock type and therefore geothermal heat flux, or both?

**P2 L5**: To the list of references *...Gow and Williamson, 1976; Paterson, 1983; Thorsteinsson et al., 1999; Weikusat et al., 2017* you could add Morgan and others (1998), which clearly shows the topographically-driven reduction in shear strain rates in the last few hundred metres of ice above the ice-bedrock interface near the summit of Law Dome where the ice thickness is $\sim 1200\,\mathrm{m}$.

**P2 L8**: I would suggest that 'strain heating' is a more widely accepted term than *shear heating*.

**P2 L7-10**: Surface meltwater that penetrates to the bed is not widespread in Antarctica. Refreeze of melt generated at the bed is probably more likely to be an issue there. So, in order for this sentence to apply equal well to both Greenland and Antarctica you could simply end at P2 L9 after *...water*.

**P2 L11-22**: In this paragraph the use of an Arrhenius relationship to describe the temperature dependence of ice strain rates is discussed. Mostly this discussion is fine; however, the paragraph requires some rewriting.

The fact that a simple Arrhenius relationship does not apply as temperatures approach the pressure melting point, unless the apparent activation energy is also a function of temperature, is discussed. Morgan (1991) acknowledge this fact, yet the actual citation for Morgan (1991) appears before this point in the discussion, leaving the impression that Morgan wasn't aware of this issue.

I'm also surprised that Budd and Jacka (1989) is not cited in this paragraph. Data from their Tables 3 & 4 and Figure 7 make it clear that a simple Arrhenius relationship – or even one with a single temperature dependent switch in activation energies, as discussed by Paterson (1994) – is inadequate to describe the effect temperature on strain rates as temperatures approach the pressure melting point. Their Table 4 is particularly important in this respect. The requirement for a temperature dependent apparent creep activation energy is also discussed by Morgan (1991). As an aside, while it's not clear – since there are no acknowledgements in Budd and Jacka (1989) – some of the data from Morgan (1991) are actually included in the compilation of Budd and Jacka (1989).

It's not discussed why the temperature dependence specified by Paterson (1994) is used in preference to Cuffey and Paterson (2010), since the latter is a revision of Paterson (1994). The justification for using Paterson (1994) should be provided. Figure 1 (next page) is something I've used in my own work, but it's useful here. It clearly shows the difference between the Paterson (1994), Cuffey and Paterson (2010) and Budd and Jacka (1989) specifications of the temperature dependence ($A(T)$ is the temperature dependent term from a Glen-type flow

relation. It is clear that the Budd and Jacka (1989) values, which are based on a compilation of experimental data, display a much higher sensitivity of strain rates to temperature as the melting point is approached. Any of these descriptions of the temperature dependence could be used with a Glen-type flow relation – the point in relation to this paper is that the Glen flow relation may do a better or worse job when simulating strain rates simply based on specification of the temperature dependence. This is discussed later in relation to section 3.3 of the manuscript.

[Figure]

Figure 1: Variation of the temperature dependent term, $A(T)$ in a Glen-type creep power law. The temperature dependence schemes of Paterson (1994), Cuffey and Paterson (2010) and Budd and Jacka (1989) are plotted. For consistency, the conversion from octahedral shear stress and strain rates to equivalent effective stresses and strain rates has been made for the data of Budd and Jacka (1989)

**P3 L15-17**: *The coarse grains and multi maxima CPO are thought to be the result of rapid strain induced boundary migration (SIBM) in combination with the nucleation of new grains (SIBM-N).* Is an adjective required here to indicate how much nucleation might be expected? If grain coarsening is significant in these layers then SIBM is likely playing a more significant role than nucleation?

**P3 L23 – P4 L3**: In this paragraph the reasons for using the flow relation of Goldsby and Kohlstedt (2001)

are well articulated, but some additional words to clarify why the Glen flow relation, with the Paterson (1994) temperature dependence, makes a meaningful 'control' would be helpful.

**P4 L8-11**: I'm not sure if I'm missing something in the following two sentences: *Ice deposited during the last glacial period (glacial ice) lies below reaching a depth of 2207 m. The layering below the glacial ice is strongly disturbed and four stratigraphic disruptions were identified at 2209.6 m, 2262.2 m, 2364.5 m and 2432.2 m of depth by discontinuities in oxygen stable isotope values...* Did you mean 'The layering in the (underlying?) glacial ice is strongly disturbed...'? As it stands *The layering below the glacial ice is strongly disturbed...* suggests there's another layer beneath the end of the (youngest) glacial ice, and that the glacial ice layer is at the most 2.5 m thick since by 2209.6 m we're into another type of ice.

**P4 L25-26**: *The total area classified as 'grain' by the Ice-image software was divided by the number of grains and a mean grain diameter was calculated assuming circular grains.* A brief comment on whether this introduces any bias into the grain size estimates would be helpful here.

**P5 L7-9**: *For each orientation images the Woodcock parameter was calculated (Woodcock, 1977). The Woodcock parameter is often used in order to distinguish between cluster and girdle type of CPOs. A distribution with a Woodcock parameter >1 indicates a cluster, while a Woodcock parameter <1 indicates a girdle.*
Woodcock (1977) actually specifies two parameters for describing the CPO pattern and strength. I'm not sure how common it is, but I haven't previously seen either referred to as a 'Woodcock parameter'....not to say that you can't, but it might be simpler and clearer to include the equation for parameter $K$, equation 5 in Woodcock (1977), since that gives you access to the relevant letter or symbol to denote the parameter.

**P5 L11-15**: *However, these microstructural parameters can differ significantly from the microstructural parameters obtained by the Ice-image software (www.ice-image.org) (Binder et al., 2013; Binder, 2014). For instance, the mean grain size derived from the orientation images in this study is systematically shifted towards lower values, which is mainly caused by the exclusion of grains with a grain diameter <0.3 mm in the LASM method as was described above.* This needs to be rewritten to make it clearer which method it is that underestimates grain size. Grain sizes determined via the LASM are an effective diameter, is this also the case for the FA measurements? From P7 L4-16 it's not clear if this is the case. A clear statement that all grain sizes reported in the manuscript are effective diameters, if that's true, would be helpful.

**P5 L28-29**: $\dot{\epsilon}_{basal}$ refers to GBS accomodated by basal slip, while $\dot{\epsilon}_{GBS}$ represents basal slip accommodated by GBS. I guess I've missed something but the difference between these two terms isn't immediately clear to me. Perhaps this could be expanded on, even if $\dot{\epsilon}_{basal}$ is discarded. Also, there's also a typo in the first 'accommodated'.

**P6 L6**: *The value for p determines whether the creep is grain size insensitive $(p = 0)$ or grain size sensitive $(p \neq 0)$.* While it is noted later in the manuscript it would be worth mentioning at this point that when $(p = 0)$ the flow relation reverts to the Glen flow relation (even if the Glen flow relation is formally introduced in the following paragraphs).

**P6 L16**: *where $\rho_{ice}$ is the density of ice $(910\,kg\,m^{-3})$.* The bulk density of ice (and snow and firn) varies considerably in ice sheets, so $910\,kg\,m^{-3}$ is a value selected (or assumed) for the present calculations, so this should be made clear. Also, is there a density profile (modelled or measured) for the NEEM core? If so, how does the chosen value relate to that profile for the range of ice depth in this study.

**P6 L29-30**: *...a constant equivalent stress of 0.07 MPa was taken as input for Glen's flow law and the composite flow law.* Even if it was explained in the companion paper, some commentary on how and why a stress of 0.07 MPa is appropriate for this study should be provided. The authors should also specify if it's the effective stress of Nye (1957) or the octahedral shear stress (e.g. Jaeger, 1962), or something else again, that's being used in the calculation.

**P7 P17-18 & Figure 1**: Based on the pole figure in Figure 1c it's not clear to me that this is a partial girdle. It appears that broad single maxima could also be a reasonable description. Either way, it's not that important

to the main focus of the paper; however, if the equations for the fabric parameters, $K$ and $C$, as defined by Woodcock (1977) were included in the manuscript, the pole figures could be annotated with these values, which would help with interpretation of the pole figure.

In the Figure 1 caption and annotations the same acronyms for the image type, as defined section 2.1 should be used, i.e. LASM and FA. These acronyms should be made consistent throughout the manuscript. While the Figure 1 caption begins with *Part of...* it should be revised to more clearly state that the presented LASM and FA images have been extracted from larger sections. Figure 1 would be better if the LASM and FA image pairs were co-registered, i.e. they showed the same region of each section (perhaps this isn't so straightforward...and that's why it wasn't done?).

**P7 L28**: *...with a Woodcock parameter varying from 0.3-3.* It looks more like 0.3-10 to me.

**P8 L4-19**: In this paragraph the contribution of stress relaxation to the microstructural evolution should be included. Irregular bedrock topography disrupts simple shear dominated flow, leading to a reduction in the stress towards the bed while at the same time the temperature of the ice is increasing (i.e. maximum simple shear strain rates occur some distance above the ice-bedrock interface). As a first estimate the extent of the flow disturbance can be related to the effective wavelength and amplitude of the bedrock topography.

While the ice temperatures have been corrected for pressure, a second order effect when it comes to the different $T^*$ at the transition for each ice core is the assumption of constant $\rho_{ice}$ as a function of depth (for the NEEM site). Is this at all significant? Possibly not, but if different values of $\rho_{ice}$ are used for each site, it may contribute to difference in $T^*$ at the transition.

**Table 1**: I'm not sure how much value is added to the manuscript by the last column of data and the last sentence of the caption. Is it really necessary?

**P8 L27-32 & Figure 3**: The last sentence of this paragraph should explicitly state that the poor performance of the Glen flow relation in predicting the temperature dependence of strain rates (if that's how you want to describe it) is directly related to selection of the Paterson (1994) apparent creep activation energies and pre-exponential terms. Comments to this effect should propagate into the Discussion and Conclusions (where appropriate). From an ice sheet modelling perspective the Paterson (1994) (or Cuffey and Paterson (2010)) temperature dependent terms are easy to implement, popular, and as described in the manuscript, overly restrictive. As shown in Figure 1 of this review, a Glen type-flow can be made to be much more sensitive to increasing temperatures according to the data used to calibrate the temperature dependent term...it's possible the data of Budd and Jacka (1989) may even lead to a greater temperature sensitivity than that shown in Fig. 3 for GBS-limited creep when $T > 262K$.

I'm not sure about referring to GBS-limited creep and dislocation creep as *end members*. As used here there are two deformation mechanism 'components' in the flow relation of Goldsby and Kohlstedt (2001), and finding a way to refer to them in this manner would have the small benefit of saving a few words here and there in the main text, but also the figure captions. Since two of the original four mechanisms from the flow relation of Goldsby and Kohlstedt (2001) are not required in this analysis, what remains is a set with two members. While technically each is an end member, describing them as end members is both redundant and suggestive of there being more than two components to the relation. I only mention this as its best to avoid any hint or suggestion that all four components of the Goldsby and Kohlstedt (2001) relation were used here.

**P9 L7-8**: *Glen's flow law predicts a higher strain rate (about $10^{-10}\,s^{-1}$) than the modified composite flow law in the lower part of the glacial ice.* Based on inspection of Fig. 3 this was always going to be the case for terrestrial ice, i.e grain size of $\sim 5mm$. A broader question is how do the calculated strain rates, for both flow

relations, compare to the expected in situ values.

**P9 L8-9**: *At the interface between the glacial ice and Eemian-glacial facies, the calculated strain rate for the composite flow law drops by about an order of magnitude.* There are some comments on localisation in the Discussion, but the question of how realistic such a rapid transition in strain rates of this magnitude is requires further attention. More on this later...

**P9 L15-18 & Figure 4**: *The strain rate produced by dislocation creep, which is not affected by the variation in grain size, steadily increases with depth throughout the Eemian-glacial facies. Glen's flow law, which is not affected by grain size variation either, predicts an increasing strain rate with depth and a higher strain rate than the modified composite flow law in the entire Eemian-glacial facies.* While this is the results section, I think some minor rewriting of these sentences to note the effect of temperature on the increase in strain rates with depth is required.

Figure 4 would benefit from an additional pane showing temperature as a function of depth for this part of the NEEM borehole - even temperature without the pressure correction would be very helpful.

**P9 L24-26**: *The difference in activation energy for Glen's flow law and the dislocation creep mechanism above their temperature thresholds is rather small (Table 2), which results in an almost similar strain rate increase with depth.* Looking at Fig. 4 I would say that these strain rate increases are not at all similar in magnitude...it's a log scale.

**P11 L26-29**: *The in-situ temperature at which coarse and interlocking grains start to appear in polar ice sheets occurs at $T^*$ of about $-11\,°C$ (262K) (Table 1), falls within the temperature range (258K to 263K) of the transition to a more temperature sensitive deformation mechanism during deformation tests (Mellor and Testa, 1969; Barnes et al., 1971; Weertman, 1983; Paterson, 1994; Goldsby and kohlstedt, 2001).* This sentence needs to be rewritten. Also, Budd and Jacka (1989) should be included in this list of references since their data compilation clearly shows the likely effect premelting has on increasing strain rates above $\sim 263\,K$.

**P12 L1-11**: This discussion, particularly the final point that *the strain rate becomes progressively more temperature dependent when approaching the melting point.* highlights why it's a good idea to calibrate the Glen flow relation with a temperature dependence, other than of Paterson (1994). Barnes and others (1971) is cited here when discussing how the apparent activation energy for creep changes markedly towards the melting point – this also shown by the data of Budd and Jacka (1989) which covers a wide range of temperatures and stresses.

**P12 L26**: *Due to the limited SIBM...* Since this hasn't been directly observed it should noted that there is an 'expectation' of limited SIBM.

**P12 L30**: Rather than *...using the available creep laws...* this should be 'chosen', or 'selected' or similar.

**P13 L5**: *Alternatively, the original microstructure might have been obliterated by SIBM.* When? In situ, or post-drilling but prior to analysis?

**P13 L10-12**: *The CPO of these coarse grained ice core sections suggests that new grains with a high Schmidt factor nucleate continuously (e.g. Alley, 1988; Montagnat et al., 2015; Qi et al., 2017). A high Schmidt factor indicates grains with a high resolved shear stress on the basal plane, i.e. grains with a soft orientation.* At some point in these two sentences it needs to be made clear that the Schmid factor for the new grains is expected to be high based on an assumed or inferred stress configuration. Also, it's Schmid, not Schmidt.

**P13 L10-12**: *This strongly suggests that the multi maxima CPO in the premelting layer is linked to the formation of new strain free grains with soft orientations (high Schmidt factor), which grow at the expense of grains oriented in a hard orientation.* Should this be 'relatively high Schmid factor'? Given that ice with both a strong single maximum and multi-maxima CPO can co-exist in regions of the ice sheet where the large-scale deformation is dominated by simple shear, it would seem appropriate to classify the Schmid factor of the multi-maxima CPO as mid-to-high range, rather than high, which would apply for a single maximum CPO.

**Section 4.5**: As a general comment this section of the paper should be revised to make it more concise. The

writing here is less focussed than elsewhere in the manuscript.

This entire section also requires revisions to make it clear that some of these interpretations, while completely valid, are speculative in the sense that they have been not verified by direct measurement. In reality this only means changing a few words here and there. For example, *This is also shown by the results using the modified composite flow law....*, might be *This is also suggested by the results using the modified composite flow law.....* Also, using the composite flow relation of Goldsby and Kohlstedt (2001) to conclude that *the fine grained ice is much softer than the coarse grained ice* isn't an independent assessment of viscosity (even if it is true), since this is answer you were always going to get using that flow relation.

**P13 L26 – P15 L24**: While the relative effects of different CPO's are discussed with regard to localisation, an omission from this section on CPO and its influence on strain rates is a clear link back to the predicted strain rates of Fig. 4. In particular, whether or not the strain rates are realistic and how they might vary if CPO effects were able to be accounted for in either of the selected flow relations should be commented on. Exclusion of CPO effects also misses the opportunity to assess the relative contribution of CPO and different mechanisms to the overall deformation story.

**P13 L26 – P14 L3**: Some care is required when discussing the strain rates measured for ice with different CPO's in the experiments of Russell-Head and Budd (1979) and Lile (1978). While the experiments of Russell-Head and Budd (1979) do show similar simple shear deformation rates for nearly-isotropic samples and those with a multi-maxima CPO, this isn't really a valid 'apples with apples comparison' due to the different in situ flow regimes of the near surface (∼isotropic) and multi-maxima (near bedrock) samples. The comparison of greater relevance to the present study, is the difference in the simple shear deformation rates of the single maxima and multi-maxima samples. This is also a more realistic comparison since the same large-scale in situ flow regime applies in each case.

In addition to (and inspired by) the work of Lile (1978) and Russell-Head and Budd (1979) the experiments of Treverrow and others (2012), Figure 8, also show how ice with a single maximum CPO deforms much more readily in simple shear than isotropic and weakly clustered samples.

**P13 L32-33**: All of the natural ice samples in Lile (1978) were from Law Dome ice cores, either from the dome summit, or a site near Cape Folger along a flow line from the summit, i.e. Lile's Dome Summit is simply the summit of Law Dome.

**P14 L2-3**: *Samples with a multi maxima CPO from Dome Summit showed a similar strain rate as laboratory-prepared isotropic samples during uniaxial compression tests (Lile, 1978).* If this is a reference to Figure 2 of Lile (1978) the sentence should be deleted since it is an incorrect interpretation. The samples from the dome summit site were from a depth of 318 m, where the total ice thickness is ∼ 1200 m. These samples have a small circle girdle CPO and are from a site where the in situ deformation regime is essentially uniaxial compression. Figure 2 of Lile (1978) clearly shows that these samples deform several times faster than isotropic ice when appropriately aligned to replicate the in situ orientation and stress configuration.

**P14 L16-17**: *while the coarse grained layers do not have a compatible CPO and are likely relatively stagnant or deform in coaxial deformation.* I think '....coarse grained layers have a (much?) less compatible CPO...' would be better.
I'm not sure what observations exist to support the claim that these layers deform coaxially. This statement should be either removed, or at least moved to the section related to borehole observations and revised.

**P14 L22**: *Since the CPO of this layer (Figure 1c) is relatively hard in simple shear.....* It's most likely still more compatible than an isotropic aggregate, so perhaps 'less compatible' would be an improvement here.

**P15 L3-5**: *Borehole data from Byrd station showed that the tilting rate (deformation rate) in the premelting layer (1810 m down to the bedrock at 2164 m of depth), where grains are coarse with a multi maxima CPO (Gow*

*and Williamson, 1976), deformed much less than the remainder of the ice (Paterson, 1983)* See Morgan and others (1997, 1998) for more of the same.

**P15 L3-5**: *Borehole data from the deeper part of the EDML ice core showed that a coarse grained layer with a girdle type CPO deformed predominantly by pure shear, while the layers just above and below deformed predominantly by simple shear (Jansen et al., 2017).* Based on my reading of the Jansen and others (2017) abstract I thought that simple shear was dominant throughout the deeper part of the core...perhaps I missed something? Nevertheless, the suggestion of direct measurements of strain rate localisation consistent with changes in CPO is particularly interesting.

Some additional details regarding these measurements at EDML are required to add weight to the discussion of strain rate localisation and the possibility of coaxial deformation, in order to make it less speculative. Without adding too much to the discussion I think it's important for readers to know about the vertical resolution of the inclination and borehole closure measurements (and any other relevant details) if they are to support the occurrence of coaxial deformation within zones where the CPO is interpreted as being compatible with this mode of deformation.
In general I understood the borehole closure problem to be a fairly tricky one since the state of stress within the ice sheet in the immediate vicinity of the borehole is modified by the existence of the borehole, so one has to deal with secondary and tertiary creep effects (depending on the timescale of the closure measurements).

**P15 L18-24**: This paragraph needs to be revised to highlight the speculative nature of the discussion. The writing has a bias towards the presentation of facts rather than speculation. Preferably an explicit statement, along the lines of the last sentence, should appear at the start of the paragraph, rather than the end.

**P16 L9-12**: *Glen's flow law, which is grain size insensitive, is unable to predict the strong variations in strain rate in the Eemian-glacial facies and predicts a steadily increasing strain rate with depth caused by an increase in temperature with depth. Glen's flow law predicts a higher strain rate than the modified composite flow law along the entire lowest 540 m of depth of the NEEM ice core.* This should be revised to address the influence of the chosen method to calibrate the temperature dependence of the Glen flow relation on its capability to predict strain rates (e.g. Figure 1 of this review).

**P16 L22-23**: *while the coarse grained interglacial layers with a partial girdle type of CPO deform at much lower strain rates by coaxial deformation* This should be revised to note the speculative nature of the suggested localised coaxial deformation.

**Technical corrections**

**Figure 1 caption, P26 L5**. Typo at *Is parallel*.

**P11 L29**: Typo at *kohlstedt*.

**P12 L32**: Missing data at *...going up to 75% at xxx m depth....*

**References**

Barnes, P., D. Tabor and J.C.F. Walker, 1971. The friction and creep of polycrystalline ice, *Proceedings of the Royal Society of London, series A*, **324**, 127–155.

Budd, W.F. and T.H. Jacka, 1989. A review of ice rheology for ice sheet modelling, *Cold Regions Science and Technology*, (16), 107–144.

Cuffey, K.M. and W.S.B. Paterson, 2010. The Physics of Glaciers, Elsevier, 4th ed.

Goldsby, D.L. and D.L. Kohlstedt, 2001. Superplasic deformation of ice: Experimental observations, *Journal of Geophysical Research*, **106**(B6), 11017–11030.

Jaeger, J.C., 1962. Elasticity, Fracture and Flow, Methuen, London, 2nd ed.

Jansen, D., I. Weikusat, T. Kleiner, F. Wilhelms, D. Dahl-Jensen, A. Frenzel, T. Binder, J. Eichler, S.H. Faria, S. Sheldon, C. Panton, S. Kipfstuhl and H. Miller, 2017. In situ-measurement of ice deformation from repeated borehole logging of the EPICA Dronning Maud Land (EDML) ice core, East Antarctica., Geophysical Research Abstracts, EGU General Assembly, vol. 19.

Lile, R.C., 1978. The effect of anisotropy on the creep of polycrystalline ice, *Journal of Glaciology*, **21**(85), 475–483.

Morgan, V.I., 1991. High-temperature ice creep tests, *Cold Regions Science and Technology*, (19), 295–300.

Morgan, V., T.D. van Ommen, A. Elcheikh and J. Li, 1998. Variations in shear deformation rate with depth at Dome Summit South, Law Dome, East Antarctica, *Annals of Glaciology*, (27), 135–139.

Morgan, V.I., C.W. Wookey, J. Li, T.D. van Ommen, W. Skinner and M.F. Fitzpatrick, 1997. Site information and initial results from deep ice drilling on Law Dome, Antarctica, *Journal of Glaciology*, **43**(143), 3–10.

Nye, J.F., 1957. The distribution of stress and velocity in glaciers and ice sheets, *Proceedings of the Royal Society of London, series A*, **239**, 113–133.

Paterson, W.S.B., 1994. The Physics of Glaciers, Pergamon, 3rd ed.

Russell-Head, D.S. and W.F. Budd, 1979. Ice-sheet flow properties derived from bore-hole shear measurements combined with ice-core studies, *Journal of Glaciology*, **24**(90), 117–130.

Treverrow, A., W.F. Budd, T.H. Jacka and R.C. Warner, 2012. The tertiary creep of polycrystalline ice: experimental evidence for stress-dependent levels of strain-rate enhancement, *Journal of Glaciology*, **58**(208), 301–314.

Woodcock, N.H., 1977. Specification of fabric shapes using an eigenvalue method, *Geological Society of America Bulletin*, **88**(9), 1231–1236.

---

## Author Comment (AC1) · 2 Sep 2019

Response to Referee #1 Dave prior
Manuscript review: tc-2018-275

**Response to general comments**

5    We would like to thank Dave Prior for the detailed and elaborate comments and suggestion on the manuscript. These were very helpful and improved the manuscript significantly. We have largely implemented the suggestions from the referee in the revised manuscript.

**Referee's first comment**

10    *Premelting*
**Authors response** We have modified the writing about premelting as suggested by the reviewer and we have also added some lines (and references) describing the indirect evidence for premelting obtained from attenuation experiments.

15    **Referee's second comment**
*A schematic overview at start*
**Authors response** A schematic overview of the structural and stratigraphic complexities has been added to the manuscript (figure 1). It shows the age of the ice, in-situ temperature, $\delta^{18}O_{ice}$ record and the stratigrapic discontinuities. Parameters like grain size and CPO were already shown in Figure 4
20    (Figure 5 in new version), so they were left out of the overview figure at the start. Related to this new figure we have also changed the descriptions of the finer-grained ice that occurs between 2207 and 2432 m of depth. In the original paper we described the finer-grained ice as glacial, however this ice is late Eemian in age according to the reconstruction of NEEM community members (2013).

25    **Referee's thrid comment**
*Grain numbers in fig 1.*
**Authors response** The second sentence of section 3.1 mentions that only a part of the 90 x 55 mm ice core section is shown, while the pole figures shows all the grain in the 90 x 55 mm ice core section.

30    **Referee's fourth comment**
*Woodcock parameter*
**Authors response** The explanation of the Woodcock parameter has been extended in Section 2.1. The equation that calculates the Woodcock parameter (*k*) from the principal eigenvalues has also been added (Equation 1)
35
**Referee's fifth comment**
*Names for the modified flow laws*
**Authors response**

40    **Referee's sixth comment**
*Eemian Glacial Facies vs. Eemian ice*
**Authors response**

**Referee's seventh comment**
45    *Strain energy during GBS*
**Authors**
We agree that the micrographs show evidence for grain boundary migration, which implies differences in internal strain energy that are probably produced by the basal slip component of deformation. Even when GBS is the rate limiting process both GBS and basal slip accommodate similar

amounts of strain as they are sequential processes.  A line has been added Page 11 line 6-8  to cover this comment.

**Referee's eighth comment**

*Split Figure 2 into 2 graphs*

**Authors response** Figure 2 (Figure 3 in the new version) has been split into two graphs.

**Referee's nineth comment**

*Fig 4.*

**Authors response** the Figure has been adjusted and is much clearer not. However, we feel that all the information in the caption is needed to interpret the figure correctly, so we decided not the change the caption of this figure.

**Referee's tenth comment**

Minor things: through manuscript

**Authors response** We have followed the suggestions from the reviewer. The change in Q concerning Glen's temperature dependency is taken from Paterson (1994). In response to reviewer 2, we have added several lines to acknowledge that the temperature dependency we have used is a simplification (Budd and Jacka, 1989).

**Referee's eleventh comment**

*"Accommodated by"*

**Authors response** Throughout the entire paper (and the companion paper tc-2018-275) we have adopted the 'rate limiting' terminology instead of the 'accommodated by' terminology. The two bullet points are incorporated in the methods now (Equation 2 and 3 in the new version).

**Referee's twelfth comment**

*The "Glen" law*

**Authors response** Similar to the companion paper (tc-2018-274) we added at the end of section 2.2 that "the form of Glen's flow law that is most often used has a stress exponent of n=3". The value of n=3 was taken from Paterson (1994) and has been cited accordingly.

**Referee's thirteenth comment**

*Girdle*

**Authors response** We describe the CPO in the coarse grained Eemian ice as a 'small circle girdle type CPO' or a 'partial girdle'. The last sentence of section 3.1 it is also mentioned that 'the c-axes are distributed in a partial girdle spanning about 30° to 40° from the vertical axis'. The pole figures in Figures 2 and 5 will help the reader further to clearify what kind of girdle type CPO is found in this coarse grained Eemian ice.

**Referee's fourteenth comment**

*CPOs during GBS in ice.*

**Authors response** we have referenced the Craw et al. paper in part 1.

**Referee's fifteenth comment**

*Figure Captions*

**Authors response** A few words were removed at some of the figure captions. However, we think that the most of the figure captions explained the figures well and therefore they were not changed or shortened any further.

---

## Author Comment (AC2) · 2 Sep 2019

Response to Referee #2 Adam Treverrow
Manuscript review: tc-2018-275

Using a composite flow law to model deformation in the NEEM deep ice core,

5    Greenland: Part 2 the role of grain size and premelting on ice deformation at high homologous temperature

Ernst-Jan N. Kuiper, Johannes H. P. de Bresser, Martyn R. Drury, Jan Eichler, Gill M. Pennock, Ilka Weikusat

10    ## Response to general comments

We thank Adam Treverrow for providing us with such elaborate and useful feedback on the manuscript. The feedback was very helpful in improving the manuscript and we implemented most of the suggestions that were given. We recognize that the quality of the writing decreased in the last part of the discussion and conclusions. We have rewritten these sections and think that the quality of

15    the writing is similar to the other parts of the manuscript now. Special attention during the rewriting of these sections was given to making sure that these sections are more concise and clearly stated the difference between interpretation of the data and speculative comments.

Below is our response to specific comments. The page and line numbers correspond to the first version of the manuscript.

20

**P1 L2, L3:** The wording in the abstract has been changed to "relatively impurity-rich and fine grained ice deposited during the glacial period" and "low impurity and much coarser grained ice deposited during the Eemian period". Throughout the manuscript we made several changes to clarify the difference in impurity content and grain size between the ice deposited in the glacial period and the

25    interglacial period.

**P1 L15-18:** We added "based on the flow relation of Goldsby and Kohlstedt (2001)" to the abstract so that it is clear that these results were obtained using the G&K flow law.

**P1 L19:** Throughout the manuscript we have replaced "impurity-depleted Eemian ice" by "low impurity Eemian ice".

30    **P2 L1:** We have changed "shallower ice" for "ice closer to the surface"

**P2 L2:** Strictly speaking we do not know the shear stress profile along an ice core. The shallow ice approximation is an approximation, which might break down at the bottom of the ice core under influence of topographic constraints. Even if the shear stress increases linearly with depth, then the strain rate (or borehole inclination) does not have to increase linearly with depth either since the

35    strain rate also depends on CPO, grain size, grain shape, etc. We feel that, as a first order approximation, the expression "shear stress increase towards the bedrock" is appropriate.

We changed 'bedrock variations' into 'variations in bedrock topography'.

**P2 L5:** Morgan et al. (1998) was added to the references.

**P2 L8:** 'shear heating' was replaced by 'strain heating'.

40    **P2 L7-10:** The sentence was ended after 'water'.

**P2 L11-12:** The citation of Morgan et al. (1991) was replaced in the paragraph.

Budd and Jacka (1989) was added to the paragraph. We have expanded the paragraph to mention that we used a simplified description for the temperature dependence of Glen's law, which is widely used in ice sheet models. As the only grain size sensitive flow laws available (Goldsby and Kohlstedt,

45    2001) use a constant activation energsy in the high temperature regime, we will follow this approach.

**P3 L15-17:** It was added that SIBM is likely more important than nucleation based on the coarse grain size.

**P3 L23 - P4 L3:** In the end of the paragraph it was added that comparing the results of the composite flow law with the results from Glen's flow law can be seen as a control.

**P4 L8-11:** For clarity we added the words "in the underlaying".

**P4 L25-26:** A sentence was added to clarify whether this introduces a bias or not.

**P5 L7-9:** The equation for the Woodcock parameter was added to the text, including some more explanation about the meaning of the Woodcock parameter (k).

**P5 L11-15:** It was added that the term 'grain size' in the remainder of this paper means 'effective diameter'.

**P5 L28-29:** The text in this sentence has been adjusted. Throughout the manuscript we also replaced the expression 'accommodated by' for 'rate limited by'.

**P6 L6:** With p≠0 the G&K flow law indeed reduces to a Glen-type flow law. However, the activation energy, pre-exponential factor and stress exponent are still different from (the most often used form) of Glen's flow law. We therefore prefer to leave this part of the text like it was.

**P6 L16:** To our knowledge there is no published density profile of the NEEM ice core. However, it is likely that the density profile along the NEEM ice core follows a similar pattern compared to other polar ice cores. Therefore, only in the upper ~100m the density of the ice deviates significantly from the 910 kg/m$^3$. The error induced by assuming a constant density of 910 kg/m$^3$ is likely much smaller than the error induced by other parameters and assumptions made in determining the stress distribution along the NEEM ice core.

**P6 L29-30:** We added some explanation for the choice of the equivalent stress level.

**P7 L17-18 & Figure 1:** We added in the paper that this could also be described as a multi-maximum CPO. The caption of Figure 1 (which became Figure 2 in the new version) has been adjusted so that LASM and FA are mentioned. The other referee of this manuscript preferred that in the results section it should be made clearer that the LASM and FA image that are shown have been extracted from the larger 90 x 55 mm ice core section. Showing the same region is technially not possible as the FA image and LASM image are not made from the same surface, but are a small distance apart (but parallel to each other).

**P7 L28:** We changed this to 0.3-10.

**P8 L4-19:** At the end of this paragraph we added a few lines that mentions the effect of stress relaxation due to the bedrock topography.

**Table 1:** We added the last column and the last sentence in the caption to show that the sudden change in CPO and grain size close to the bedrock does not always coincide with a transition from glacial ice to interglacial ice or vice versa. We think this shows that the sudden change in microstructure is not only caused by glacial or interglacial ice, but that a temperature 'threshold' also plays a role in this sudden change. We therefore prefer to leave the Table and the caption as it is.

**P8 L27-32 & Figure 3:** We have added a sentence to acknowledge that different versions of Glen's law could be used with higher temperature sensitivity, So the results shown what is now figure 4, are dependent on our use of the Paterson (1994) version of Glen's law.

We agree that the term 'end member' can lead to confusing or give the impression that these are the only members in the G&K flow law. We therefore changed 'end member' to 'member'.

**P9 L8-9:** We have changed the discussion of this result in the Discussion.

**P9 L15-18 & Figure 4:** We've added that the strain rate increase coincides with the increase in temperature.

The temperature profile along the NEEM ice core is shown in the companion paper (Figure 1). Since Figure 5 ((Figure 4 in previous version) is already quite full, we prefer to leave the figure as it is.

**P9 L24-26:** It was added that the similar increase is a similar increase in order of magnitude. We also added a sentence to say that the relative changes depend on the version of Glen's law that is used.

**P11 L26-29:** Budd and Janka (1989) was added to the list.

Sentence has been re-written to improve clarity.

**P12 L1-11:** Budd and Jacka (1989) was added to the paragraph.

**P12 L26:** It was added that the SIBM rate was expected to be lower in these layers.

**P12 L30:** 'Using the available creep laws' was replaced by 'using the chosen creep flow laws'.

**P13 L5:** "In situ"was added to the end of the sentence.

5   **P13 L10-12:** "Schmidt" was replaced by "Schmid".

It was added that this was based on the assumption that simple shear is the assumed sress configuration.

**P13 L10-12:** Both suggestions were implemented.

**Section 4.5:** This section of the paper has been rewritten.

10   **P13 L26 – P15 L24:** This part of the paper has been rewritten to consider the effect of CPO on the predicted strain rates.

**P13 L26 – P14 L3:** The comparison of the strain rates for ice with different CPO's has been made to argue that ice with a multi maxima CPO has similar strain rates to ice with an isotropic CPO when deformed in simple shear. This similarity suggests that the flow laws obtained for isotropic ice are

15   valid for ice with a multi maxima CPO when deformed in simple shear.

**P13 L32-33:** We have corrected this mistake.

**P14 L2-3:** This interpretation was indeed incorrect and the sentence was deleted.

**P14 L16-17:** The discussion sections have been rewritten and this sentence has been taken out of the discussion.

20   **P14 L22:** The discussion sections have been rewritten and this sentence has been taken out of the discussion.

**P15 L3-5:** We thank the referee for this suggestion and added the Law Dome example.

**P15 L3-5:** The interpretation of borehole logging data in terms of deformation mode has been removed from the discussion. As noted by the reviewer, there are complications with the

25   interpretation and not all the examples that were quoted have been fully published.

**P15 L18-24:** This paragraph is rewritten to highlight the speculative nature in the beginning of the paragraph.

**P16 L9-12:** This part has been revised to mention that the strain rate predicted from Glen's law is dependent on the choice of the temperature sensity in the flow law.

30   **P16 L22-23:** Has been revised by separating the discussion about the strainrates from the discussion of the deformation mode.

---

## Referee Report (RR1)

**Manuscript review: tc-2018-275**

Using a composite flow law to model deformation in the NEEM deep ice core, Greenland: Part 2 the role of grain size and premelting on ice deformation at high homologous temperature

Ernst-Jan N. Kuiper, Johannes H. P. de Bresser, Martyn R. Drury, Jan Eichler, Gill M. Pennock, Ilka Weikusat

**Reviewer:** Adam Treverrow**

**General comments**

The authors have made a thorough response to the reviews and the corresponding changes to the manuscript have improved it greatly. It should be published subject to the minor issues described below being addressed. As an aside, I was appreciative of the colour-coded formatting of the revised manuscript and accompanying response documents. They made navigating through the updated manuscript quite simple.

Lastly, I apologise for this review being overdue.

**Specific comments**

P1 L13, P2 L23, P9 (Section 4.2): Both 'liquid-like' and 'water-like' are used throughout the manuscript to describe the occurrence of liquid water at grain boundaries and at impurity interfaces. These should all be liquid-like since water-like is ambiguous.

**P4 L34**: Additional detail required.  $\lambda_1, \lambda_2, \lambda_3$  are normalised eigenvalues of the 2nd order order orientation tensor.

Elsewhere the expression 'c-axes eigenvalue' is used, when specific reference is being made to  $\lambda_3$ . This should be corrected.

**P5 L26**: A further relevant citation is Greve and others (2014).

**P6 L3**: Recently Bons and others (2018) have pointed out that the experimental basis for n = 3 is debatable. I understand and agree with the intent of this sentence, but as it's incorrect as written and should be revised.

Over a limited range of stresses there is a very good experimental basis for a creep power law where n = 3 – provided you're only interested in predicting secondary creep rates for initially isotropic ice. Since this clearly is not the case for the vast majority of ice deformation in polar ice sheets, it's then true that using n = 3 probably isn't the best idea for an ice sheet model, even if there's historical inertia associated with it's usage.

So while some folks would consider that selection of the flow relation stress exponent is a solved problem, there's clearly evidence suggesting that  $n \neq 3$ . The actual problem is what value to use in its place - there's not a lot of data around. From our analysis of new and existing experimental data in Treverrow and others (2012) we speculate that n = 3.5 may be an improvement...our experimental studies exploring this effect are ongoing.

So, the P6 L3 sentence needs some minimal revision to suggest that n = 3 is not appropriate for the type of high-strain (tertiary) creep that is characteristic of polar ice masses. Since the sentence describes the existence of experimental evidence for  $n \neq 3$ , you should probably explicitly cite some of these, e.g. Goldsby and Kohlstedt (2001) and Treverrow and others (2012)...there are others too.

**P6 L25** typo at 'delta O18' – elsewhere  $\delta^{18}$ O is used.

Figure 1: There are no axis labels indicating the range of  $\delta^{18}$ O values.

**Table 1**:From the author's response: Table 1: We added the last column and the last sentence in the caption to show that the sudden change in CPO and grain size close to the bedrock does not always coincide with a transition from glacial ice to interglacial ice or vice versa. We think this shows that the sudden change in microstructure is

not only caused by glacial or interglacial ice, but that a temperature 'threshold' also plays a role in this sudden change. We therefore prefer to leave the Table and the caption as it is.

I agree – the motivation for including the age data in the table is completely valid. However, getting back to my initial comments, what I should have said is that expressing the age in terms of MIS makes this information rather impenetrable to those readers who will need to seek out the corresponding reference to decode the marine isotope stages and interpret this data column.

**P10 L25**: A minor point – the discussion in this paragraph would be much clearer if the following sentence was slightly altered.

Therefore, it is argued that the temperature threshold should have been 261.7K instead of 258K

Perhaps something like: 'Therefore, it is argued that the corresponding temperature threshold should have been 261.7K instead of 258K, if a confining pressure of 50 MPa is also assumed'.

Table 3: There should be a citation to Goldsby and Kohlstedt (2001), since that's where the n = 1.8 and n = 4 values originate. It's also the basis for values of the pre-exponential term and activation energy for the GBS-limited and dislocation creep.

Section 4.5 The text in this section should be consistent. At P11 L40 it is noted that:

CPO is thought to strongly influence strain rates,

while at P12 L11 there is:

it is well known that the CPO has a weakening effect on ice depending on its orientation.

The latter is the more correct statement, since there is evidence for this, so P11 L40 should be changed accordingly. Also, in both cases the same citations are used, (e.g. Alley, 1988; Hudleston, 2015), so the statements can't be conflicting. And finally (a minor point) these references are probably acceptable here, but personally I think that Budd and Jacka (1989) and/or Faria and others (2014), would be better placed here than Alley (1988).

**P12 L15**: Here (and elsewhere) I found usage of 'premelting layer' confusing. This is because premelt is discussed over various spatial scales throughout the manuscript. There is premelt at grain boundaries and impurity sites, but also a zone within the ice, where due to temperature, premelt may be occurring. Overall, the discussion would be improved if terminology such as the temperate layer where premelt may occur\* was used when discussing the large-scale zone where this may be important.

\* OK - this example is rather verbose. You'll find your own way to describe this region within the ice sheet, which also clarifies the issue of scale.

**P12 L25-32**: The last sentence of this paragraph narrowly misses an opportunity to make an important point that's relevant to this discussion. The Law Dome DSS record is your friend here.

The DSS strain rate profile is adequately described in the manuscript. There is a broad maximum in the shear strain rate at  $\sim 1000$  m and then with increasing depth the strain rate begins to decrease, even though the ice is getting warmer. As described, this is due to a large-scale reduction in stress. Within this zone, where coarse-grained ice with a multi-maxima CPO predominates, there is narrow band where strain rates are high. This is correctly identified in the manuscript as ice from the last glacial maximum (LGM). What's important to this manuscript is that relative to the ice immediately above and below this LGM layer (where stresses and temperature are otherwise similar) significantly higher strain rates occur. These must be associated with the fine grain size and significantly different CPO of that layer. The fact that fine grained ice with a strong single-maximum CPO exists at this ice depth is most likely a consequence of the manner in which impurities influence the rate, or way in which microstructure can evolve. Immediately above and below the LGM spike there is no longer any sign of this strong CPO and fine grain size. Within the LGM layer, the single maximum and fine grain size are likely remnants of a microstructure that developed upstream of the borehole site when this ice passed though the zone where shear rates are generally higher.

So, getting back to the sentence at P12-L31-32, that the highest simple shear strain rates over the entire depth profile occur in a zone where the strain rates are otherwise decreasing points to the combined influence of temperature and grain size on strain rates. Importantly, recall that the stresses are lower in this zone than at  $\sim 1000$  m where the dominant, yet broad maximum in simple shear strain rates occurs.

**References**

Alley, R. B., 1988. Fabrics in Polar Ice Sheets: Development and Prediction, Science, 240(4851), 493–495.

- Bons, P. D., T. Kleiner, M.-G. Llorens, D. J. Prior, T. Sachau, I. Weikusat and D. Jansen, 2018. Greenland Ice Sheet – Higher non-linearity of ice flow significantly reduces estimated basal motion, *Geophysical Research Letters*, **0**(ja).
- Budd, W.F. and T.H. Jacka, 1989. A review of ice rheology for ice sheet modelling, Cold Regions Science and Technology, (16), 107–144.
- Faria, S. H., I. Weikusat and N. Azuma, 2014. The microstructure of polar ice. Part II: State of the art, Journal of Structural Geology, 61(0), 21 – 49.
- Goldsby, D.L. and D.L. Kohlstedt, 2001. Superplasic deformation of ice: Experimental observations, Journal of Geophysical Research, 106(B6), 11017–11030.
- Greve, R., T. Zwinger and Y. Gong, 2014. On the pressure dependence of the rate factor in Glen's flow law, Journal of Glaciology, 60(220), 397–398.
- Hudleston, P.J., 2015. Structures and fabrics in glacial ice: A review, Journal of Structural Geology, 81, 1 27.
- Treverrow, A., W.F. Budd, T.H. Jacka and R.C. Warner, 2012. The tertiary creep of polycrystalline ice: experimental evidence for stress-dependent levels of strain-rate enhancement, *Journal of Glaciology*, **58**(208), 301–314.

---

## Author Response (AR2)

**Response to technical corrections**

5    We would like to thank Dave Prior for the technical corrections. All the suggestions that Referee #1 made were implemented and it helped to make the manuscript more accurate and concise. The corrections made for Referee #1 and #2 are colored in red in the manuscript.

Response to Referee #2 Adam Treverrow
Manuscript review: tc-2018-275

**Response to technical corrections**

5    We thank Adam Treverrow for providing us again with very valuable feedback. All comments were included in the revised version of the manuscript.

Below is our response to specific comments. The page and line numbers correspond to the first version of the manuscript.

10    **Specific comments**

**P1 L13, P2 L23, P9 (Section 4.2):** We agree with the referee. Throughout the manuscript we have replaced 'water-like' for 'liquid-like'.

**P4 L34:** We clarified in P4 L34 that it's the second order orientation tensor. We also clarified in section 3.2 that with 'c-axes eigenvalue' we mean $\lambda_3$

15    **P5 L26:** This reference was added to P5 L26 and to the reference list.

**P6 L3:** We agree that the way it was written down here was incorrect. Glen's flow law (with n=3) is correct, but only under a certain set of circumstances (initially isotropic ice at secondary creep), which might not be the most relevant conditions for ice sheets. We have rewritten this part of the paragraph.

20    **P6 L25:** We replaced delta O18 for $\delta^{18}O_{ice}$.

**Figure 1:** A axis label was added to the $\delta^{18}O_{ice}$ axis.

**Table 1:** We agree that expressing the age of the ice at the transition in Marine Isotope Stages (MIS) can make it harder to interpret the data for someone who is not familiar with this terminology. We added the age (in ka BP) in the table as well.

25    **P10 L25:** This sentence was adjusted like Referee #2 proposed it.

**Table 3:** We added the reference of G&K (1997, 2001) and mentioned that some of the parameters remained the same.

**Section 4.5:** P11 L40 is changed as suggested by the referee.

Alley (1988) was replaced with Budd and Jacka (1989) and Faria et al. (2014b).

30    **P12 L15:** We have changed the terminology into 'premelting zone' to describe the deeper ice where the temperatures are high enough for premelting to occur. We did not use the term temperate since this implies bulk melting of the ice.

**P12 L25-32:** We expanded this sentence as suggested by the reviewer.